# Fleshy red algae mats act as temporary reservoirs for sessile invertebrate biodiversity

Yusuf C. El-Khaled [1✉], Nauras Daraghmeh[1,2], Arjen Tilstra[1], Florian Roth[3,4], Markus Huettel[5], Felix I. Rossbach [1], Edoardo Casoli [6], Anna Koester[1], Milan Beck[1], Raïssa Meyer[1], Julia Plewka[1], Neele Schmidt [1], Lisa Winkelgrund[1], Benedikt Merk [1] & Christian Wild[1]

Many coastal ecosystems, such as coral reefs and seagrass meadows, currently experience overgrowth by fleshy algae due to the interplay of local and global stressors. This is usually accompanied by strong decreases in habitat complexity and biodiversity. Recently, persistent, mat-forming fleshy red algae, previously described for the Black Sea and several Atlantic locations, have also been observed in the Mediterranean. These several centimetre high mats may displace seagrass meadows and invertebrate communities, potentially causing a substantial loss of associated biodiversity. We show that the sessile invertebrate biodiversity in these red algae mats is high and exceeds that of neighbouring seagrass meadows. Comparative biodiversity indices were similar to or higher than those recently described for calcifying green algae habitats and biodiversity hotspots like coral reefs or mangrove forests. Our findings suggest that fleshy red algae mats can act as alternative habitats and temporary sessile invertebrate biodiversity reservoirs in times of environmental change.

[1] Marine Ecology Department, Faculty of Biology and Chemistry, University of Bremen, 28359 Bremen, Germany. [2] Red Sea Research Center, King Abdullah University of Science and Technology (KAUST), Thuwal 23955, Kingdom of Saudi Arabia. [3] Baltic Sea Centre, Stockholm University, 10691 Stockholm, Sweden. [4] Faculty of Biological and Environmental Sciences, Tvärminne Zoological Station, University of Helsinki, 00014 Helsinki, Finland. [5] Department of Earth, Ocean and Atmospheric Science, Florida State University, Tallahassee, FL 32306-4520, USA. [6] Department of Environmental Biology, Sapienza University of Rome, 00185 Rome, Italy. ✉email: yek2012@uni-bremen.de

Sessile plants and invertebrates play a central role in shaping biotic communities by increasing both the structural and habitat complexity, thus, promoting biodiversity[1–3]. In the marine environment, ecosystem engineers are responsible for forming biodiversity hotspots (i.e., areas rich in rare, threatened species)[4], such as seagrass meadows[5,6], tropical coral reefs[3], and mangrove forests[7]. Ecosystem engineers in these habitats change the abiotic and biotic components of the ecosystem, and in doing so, generate structurally complex environments that benefit both the engineers themselves and the associated biodiversity[1,8]. In the Anthropocene[9], human activity has negatively impacted almost all marine ecosystems. These threats have evoked ecosystem responses[10] leading them down a path of degradation[11]. Anthropogenic stressors occurring either singularly or in combination, such as ocean warming[6] and acidification[12] or nutrient pollution[6], can alter the community dynamics, shifting the system to alternative states dominated by more tolerant species[6,12]. These transitions, e.g., shifts from the reef or hard-bottom communities towards persistent, fleshy, non-calcifying (macro-) algal assemblages, are referred to as 'phase-shifts' to alternative states[13]. Phase shifts naturally entail a series of consequences on multiple levels, such as a loss of structural/spatial complexity, a loss of ecosystem services and functioning[11,14], and consequently, a loss of biodiversity[3,6,15,16]. Identifying potential biodiversity refugia that are pivotal for rebuilding marine life[17] is therefore essential to appropriately adapt conservation strategies in times of increased biodiversity loss associated with anthropogenic global change[11,12,18] and direct local human impacts (e.g., pollution, coastal development)[5,6,19].

In the Mediterranean Sea, rocky hard-bottom communities and commonly identified biodiversity hotspots such as seagrass meadows are declining primarily due to environmental pressures[5,6,19,20]. Meadows formed by *Posidonia oceanica* seagrass rank amongst the most valuable coastal ecosystems worldwide as they provide a range of goods and ecosystem services[21,22], e.g., they exhibit high biodiversity, function as ecosystem engineers, and can act as natural coastal protection barriers[23]. *P. oceanica* meadows consist of the rhizome layer (often up to several m thick)[24] and the leaf canopy. The meadows occur from shallow waters down to depths of 40 m (depending on water turbidity). Due to anthropogenically induced environmental stressors[6], such as nutrient and sediment pollution, habitat loss and degradation[19], pollution[5,19], eutrophication[5,19] and/or ocean warming[19], seagrass meadows are among the most threatened ecosystems worldwide[25].

In parallel, these stressors could have promoted the formation of persistent[26], turf- and mat-forming algal assemblages of the species *Phyllophora crispa* (formerly *P. nervosa*[27]) that have been observed across the Mediterranean[28,29], the Black Sea[30,31] and the Atlantic[29,32]. A growing number of publications addressing *P. crispa* suggests an increase of these algae in the Mediterranean[27,28,33,34]: *P. crispa* has been observed along the coast of Sardinia, Italy[28] and lately in the Tyrrhenian Sea, Italy, for the first time[27], where it has been found in dense mats of up to 15 cm thickness (Fig. 1a). *P. crispa* is a perennial rhodophyte of the order Gigartinales that typically produces branched thalli of up to 15 cm in length[26,30]. These red algae mats tolerate large variations in key environmental parameters and can proliferate under low water temperature (<10 °C) and salinity (18 PSU)[30]. *P. crispa* is sciaphilic[27,31], i.e., adapted to low-light conditions, and reaches large accumulations in water depths between 10 and 55 m[30,31]. The thalli of *P. crispa* can exhibit either an attached growth form covering hard substrates, an unattached form growing on sediments[31], or on reefs engineered by invertebrates, as recently observed in the Black Sea[31].

Algal assemblages can support high biodiversity, with several studies having found associations between high biodiversity and drifting algae in a lagoon off the west coast of the United States[35] and in the Baltic Sea[36–38]. Furthermore, the same has been found with calcifying green algae communities in coral reefs of the Great Barrier Reef, Australia[39], green algal blooms in the United States[40], Canada[41] and South Africa[42], as well as at further locations in the Atlantic[43,44]. In addition, kelp-forming brown algae in the United States[45] and United Kingdom[46] host a vast array of associated organisms. It remains unknown, however, whether the mat-forming red alga *P. crispa*, which is increasing in abundance and potentially replacing classical high biodiversity habitats also harbours high associated sessile biodiversity. Based on recent pilot studies that have identified non-colonial[27] and sessile polychaetes[34] to be associated with *P. crispa* mats, we here determined the role of *P. crispa* as habitat for overall sessile invertebrate biodiversity. The present study aims to answer the following research questions: (i) to what extent can *P. crispa* mats function as habitat for sessile invertebrates, and (ii) how does this biodiversity compare to neighbouring *Posidonia oceanica* seagrass meadows? We focussed on the sessile biodiversity in *P. crispa* mats and adjacent *P. oceanica* meadows for several reasons. Firstly, previous pilot studies[27,34] lead to the hypothesis of a high associated sessile invertebrate diversity in *P. crispa* that is comparable in terms of community composition, invertebrate species richness and abundance to that of *P. oceanica*. This invertebrate diversity is likely linked to different habitat characteristics such as micro-niches caused by varying influences on key environmental parameters and ecosystem engineering functions[33]. Secondly, the presence of sessile invertebrates potentially reflects the stability and longevity of the red algae mats as habitats[47]. Hence, a particular sampling procedure was chosen to ensure the complete retrieval of sessile invertebrates.

## Results and discussion

**Fleshy red algae mats as biodiversity hotspots for sessile invertebrates.** We assessed the sessile invertebrate biodiversity in neighbouring *P. crispa* and *P. oceanica* habitats along the north-eastern and north-western coasts of Giglio Island, within the Tuscan Archipelago National park, Tyrrhenian Sea, Italy (see Supplementary Fig. S1). *P. oceanica* community assessments included analysis of the holobiont (leaves + subsurface structures), as well as separate analyses of the leaves and rhizomes to account for potential differences[48] (see Methods). Briefly, invertebrates were determined to the lowest possible taxonomic level. However, in case no clear identification was possible, individuals were distinguished based on distinct visual characteristics, resulting in the identification of distinct phenotypes rather than species.

We recorded 312 distinct sessile invertebrate phenotypes (covering 9 higher taxa) for both *P. crispa* and *P. oceanica*, of which 223 occurred in *P. crispa* mats and 179 in *P. oceanica* holobionts, respectively (Fig. 2a). All (sub-) habitats accommodated distinct communities (Fig. 2b), with 133, 21 and 18 phenotypes uniquely found in *P. crispa* mats, *P. oceanica* leaves and *P. oceanica* rhizomes, respectively (Fig. 2a). Approximately 25% more phenotypes were found in *P. crispa* mats than in the neighbouring *P. oceanica* seagrass meadow holobionts. Calculations of classical diversity indices further endorsed *P. crispa* as a hotspot of sessile invertebrate diversity comparable to traditional biodiversity hotspots such as coral or Mediterranean coralligenous reefs (Table 1).

The calculated abundances (mean number of individuals (ind) habitat $m^{-2}$ ± standard error; note: colonies of colonial species are considered as individuals for readability hereafter) suggest that *P. crispa* mats provide a valuable habitat for sessile invertebrates that depend on a solid surface for attachment. Our data showed 64,008 ± 4609 ind $m^{-2}$ associated with *P. crispa* mats, which was three times more than in *P. oceanica* holobionts

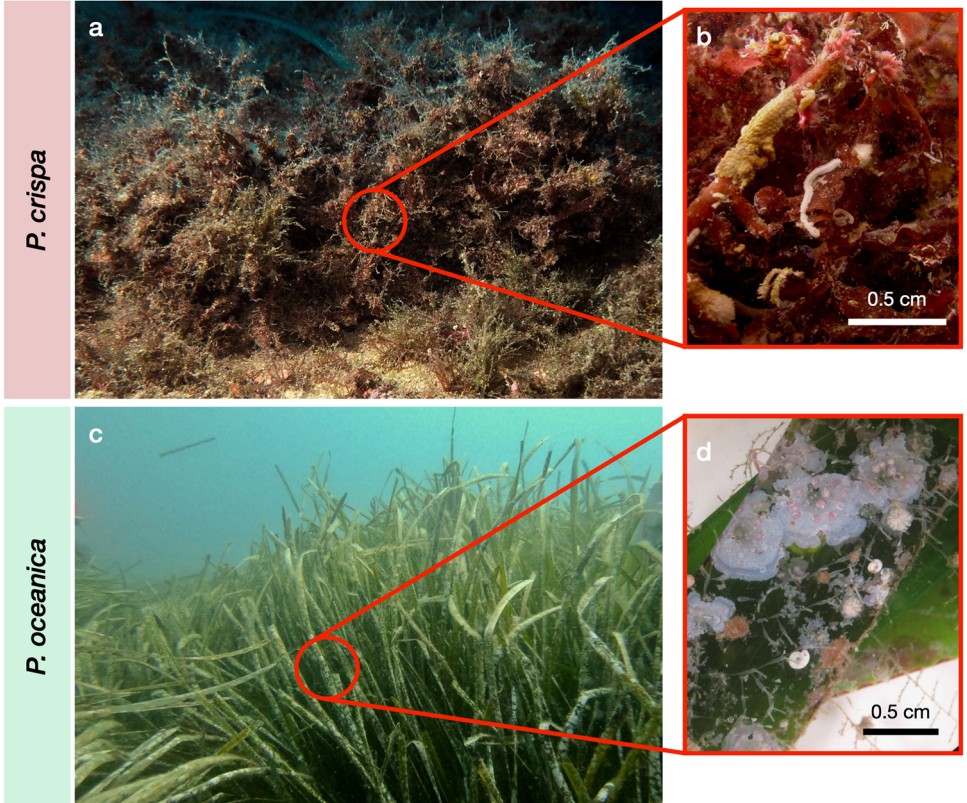

**Fig. 1 *Phyllophora crispa* mat and *Posidonia oceanica* seagrass meadow with associated sessile invertebrates.** *P. crispa* mat (**a**) and *P. oceanica* meadow (**c**) with Bryozoa, Polychaeta and Foraminifera on *P. crispa* thalli (**b**), Bryozoa, Polychaeta and crustose coralline algae (Corallinales) as epiphytes on *P. oceanica* leaves (**d**). Pictures taken by Felix I. Rossbach (**a**, **b**, **d**) and Friederike Peiffer (**c**).

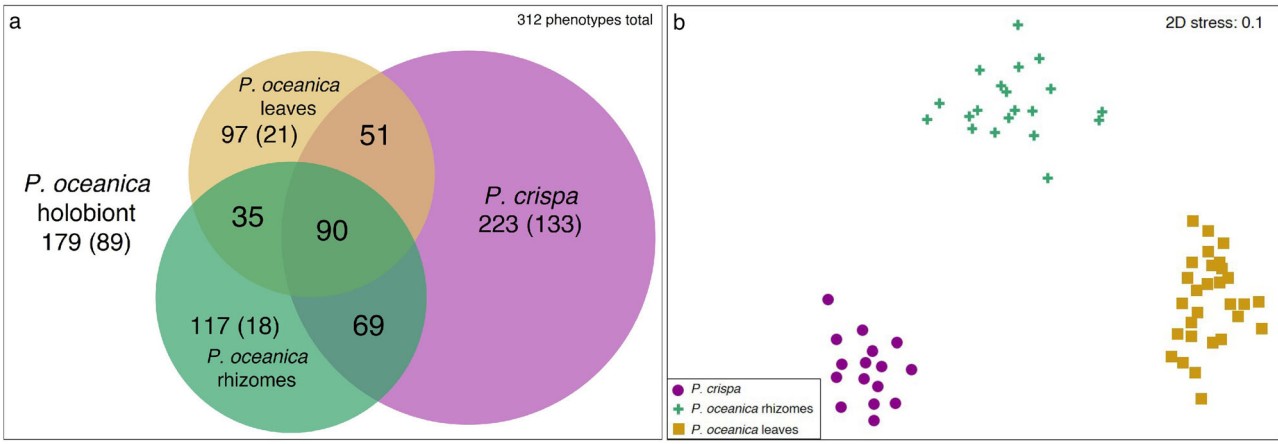

**Fig. 2 Area-proportional Venn diagram and ordination of biodiversity data by non-metric multidimensional scaling (nMDS).** Area-proportional Venn diagram (**a**) displaying numbers of total (= present in the respective habitat), shared, and unique (in brackets) phenotypes found in investigated *Phyllophora crispa* (purple), *Posidonia oceanica* holobiont, *P. oceanica* leaves (gold) and *P. oceanica* rhizomes (green); area in proportion to number of phenotypes in *P. crispa*. Ordination of biodiversity (incidence) data by nMDS (**b**) based on Bray–Curtis similarities of *P. crispa* (purple dots), *P. oceanica* rhizomes (green crosses) and *P. oceanica* leaves (gold rectangles).

(19,535 ± 1421; Dunn's test $p < 0.001$; Supplementary Table S2), four times more compared to *P. oceanica* leaves (15,857 ± 1654; Dunn's test $p < 0.001$; Supplementary Table S2) and two times the number observed in *P. oceanica* rhizomes (24,867 ± 1991; Dunn's test $p < 0.001$; Supplementary Table S2). Whereas *P. crispa* mats harboured an outstanding abundance of Bryozoa (44,222 ind habitat $m^{-2}$), both Bryozoa and Foraminifera were equally abundant in *P. oceanica* leaves and rhizomes (Supplementary Table S1). *P. crispa* harboured a similar number of phenotypes of

Bryozoa and Foraminifera (76 and 81, respectively), whereas the number of bryozoan phenotypes exceeded that of Foraminifera in *P. ocean*ica (78 and 52, respectively). In addition, we identified three distinct communities using non-metric multidimensional scaling (nMDS, Fig. 2b). The nMDS plot and appendant statistical analysis revealed that sessile invertebrate communities significantly varied among habitats (PERMANOVA with all $p < 0.001$; Supplementary Table S3), independent of the number of phenotypes and individuals of the investigated habitats.

**Table 1 Diversity indices (richness = number of sessile phenotypes, H' = Shannon, D = Simpson) and evenness accounting for sessile invertebrates for investigated as well as reference biodiversity hotspots based on literature data.**

| Habitat | Location | Richness | Taxa | Evenness | H' | D | Reference |
|---|---|---|---|---|---|---|---|
| *Phyllophora crispa* | NW Mediterranean | 223 | 9[a,b,c,e,f,m,p,r,s] | 0.6969 | 2.209 | 0.2693 | Present study |
| *Posidonia oceanica* | NW Mediterranean | 179 | 7[a,b,c,f,m,p,s] | 0.7581 | 2.128 | 0.2900 | Present study |
| *Posidonia oceanica* | S Mediterranean | 33 | 5[a,b,f,p,s] | 0.8706 | 2.021 | 0.2519 | Mabrouk et al. (2014)[106] |
| Coralligenous reefs | NW Mediterranean | 55 | 6[a,b,c,f,p,s] | 0.8070 | 2.086 | 0.2539 | Verdura et al. (2019)[107] |
| Coralligenous reefs | Mediterranean | 786[t] | 7[a,b,c,f,m,p,s] | 0.9418 | 2.644 | 0.1731 | Ballestros (2006)[108] |
| *Cystoseira zosteroides* | NW Mediterranean | 78 | 6[a,b,c,f,p,s] | 0.7574 | 1.958 | 0.3004 | Ballestros et al. (2009)[109] |
| Coral reef | SW Indian Ocean | 457 | 5[a,c,f,m,s] | 0.8765 | 2.035 | 0.2789 | Cleary et al. (2016)[110] |
| Coral reef turf algae | W Indian Ocean | 48[u] | 2[p,m] | 0.9950 | 0.995 | 0.4929 | Milne and Griffiths (2014)[111] |
| Coldwater coral reef | N Atlantic Ocean | 213 | 7[a,b,c,f,m,p,s] | 0.9523 | 2.673 | 0.1653 | Mortensen and Fossa (2006)[112] |
| Coldwater coral reef | N Atlantic Ocean | 77 | 4[a,b,c,s] | 0.8062 | 1.612 | 0.3585 | Henry et al. (2010)[113] |
| Mangrove forest | Caribbean Sea | 54 | 6[a,b,c,m,p,s] | 0.7494 | 1.937 | 0.2970 | Farnsworth and Ellison (1996)[114] |
| Kelp forest | NE Pacific Ocean | 79[v] | 6[a,b,c,m,p,s] | 0.9456 | 2.444 | 0.1912 | Graham (2004)[115] |
| Antarctic hard bottom | Weddell Sea | 608[w] | 6[a,b,c,f,m,s] | 0.8500 | 2.197 | 0.2803 | Gutt et al. (2000)[116] |
| *Halimeda* bioherm | Coral Sea | 474[w] | 5[a,b,c,m,s] | 0.6965 | 1.617 | 0.4202 | McNeil et al. (2021)[39] |

Indices and evenness presented here were calculated based on classical formulas and not based on Hill-number calculations to enable comparison with literature data (see Methods).
[a]Ascidiacea, [b]Bryozoa, [c]Cnidaria, [e]Entoprocta, [f]Foraminifera, [m]Mollusca (Bivalvia), [p]Polychaeta (Sedentaria), [r]Rotifera, [s]Porifera.
[t]Data collated from multiple other publications.
[u]Excluded Cnidaria, Bryozoa and Ascidiacea from the analysis.
[v]Respective study included barnacles and phoronids that were not included in the current analysis.
[w]Excluded Polychaeta from the analysis.

To assess *P. crispa*'s role as a potential sessile invertebrate biodiversity hotspot compared to neighbouring *P. oceanica* meadows, we performed a diversity analysis based on the concept of Hill numbers. Hill numbers account for differences in sampling efforts, i.e., number of samples collected per habitat. The resulting metric represents the effective number of equally abundant species $^{q}D$[49,50], where $q$ denotes the diversity order of a Hill number. The parameter $q$ determines the sensitivity to species' frequencies and Hill numbers based on increasing values of $q$ place more emphasis on frequently occurring species. In our analysis, $^{q}D$ of orders $q = 0$, $q = 1$, and $q = 2$ were calculated, representing phenotype richness (i.e., phenotypes quantified equally disregarding frequency, $^{0}D$), Shannon diversity (i.e., effective number of frequent phenotypes, $^{1}D$) and Simpson diversity (i.e., effective number of highly frequent phenotypes, $^{2}D$), respectively[51] (see Methods for further details).

The estimated sample completeness (i.e., diversity detected) profiles implied that there was undetected diversity within the habitats (Fig. 3a). Sample completeness profiles revealed that between 73.0% (*P. oceanica* holobiont) and 85.7% (*P. crispa*) of phenotype richness ($q = 0$) was detected with no significant differences among (sub-) habitats (i.e., respective 95% confidence intervals overlapped). The diversity detected in the (sub-) habitats rose with order $q$ (i.e., diversity detected increased for more frequently occurring species) and increasingly aligned in all habitats for Shannon ($q = 1$) and Simpson ($q = 2$) diversity, with the majority of frequent and highly frequent phenotypes being detected (Fig. 3a and Supplementary Table S6). To test if we could estimate diversity based on our data reliably, sample-size-based rarefaction and extrapolation curves were computed to check for asymptoted values of $q$. An estimation of true Simpson diversity based on our data for all (sub-) habitats was indeed reliable (i.e., size-based rarefaction and extrapolation curves asymptoted for $q = 2$; Fig. 3b). Hence, we could confirm that *P. crispa* harboured assemblages with significantly higher Simpson diversity (~132; Fig. 3c and Supplementary Table S6; no overlap of 95% confidence intervals[52]) compared to all other (sub-) habitats, which underlines its role as a biodiversity hotspot for sessile invertebrates. For phenotype richness and Shannon diversity, only conservative minimum estimates could be obtained, as size-based rarefaction and extrapolation curves did not asymptote for

$q = 0,1$ (Fig. 3b). In this case, a statistically reliable comparison between habitats' phenotype richness and Shannon diversity may only be performed based on standardised data. For this purpose, we compared diversities based on standardised data at a sample coverage level of $C_{max} = 96.9\%$ (Fig. 3d and Supplementary Table S6). $C_{max}$ is the lowest sample completeness at $q = 1$ of any (sub-) habitat when samples are extrapolated to double the respective number of samples per (sub-) habitat. Consequently, we showed that *P. crispa* exhibited significantly (i.e., no overlap of respective 95% confidence intervals[52]; Fig. 3d and Supplementary Table S6) higher phenotype richness compared to neighbouring *P. oceanica*: phenotype richness of *P. crispa* mats (~234 phenotypes) exceeded those of *P. oceanica* rhizomes (~142) and leaves (~102) at a fixed sample coverage of $C_{max} = 96.9\%$ (Fig. 3d and Supplementary Table S6[52]), whereas the difference compared to the *P. oceanica* holobiont (~207) was marginal. For Shannon diversity, *P. crispa* showed a significantly higher index value (~159) compared to the *P. oceanica* holobiont (~111), leaves (~64) and rhizomes (~84; see Fig. 3d and Supplementary Table S6[52]). Phenotype evenness (i.e., Pielou's $J$ at $C_{max}$; an evenness measure based on phenotype occurrences) was high for all habitats, being lowest for the *P. oceanica* holobiont (0.88) and highest for *P. crispa* (0.93; Supplementary Table S6). Furthermore, *P. crispa* harboured the most evenly diverse biotic communities among all (sub-) habitats for all orders of $q > 0$ at $C_{max}$ (i.e., for orders of increasing sensitivity to phenotype frequencies; Fig. 3e and Supplementary Table S6). The difference in estimated phenotype Simpson diversity between *P. crispa* and the *P. oceanica* holobiont at $C_{max}$ was larger than the difference in phenotype richness (i.e., ~50 and ~27, respectively; Supplementary Table S6). When comparing the empirical richness values (i.e., values for $q = 0$) with the values estimated asymptotically and non-asymptotically (the latter standardised for $C_{max}$), the number of undetected phenotypes was larger for (sub-) habitats of *P. oceanica* (holobiont and rhizomes) than for *P. crispa* (Supplementary Table S6). These findings indicate that the higher overall diversity in *P. crispa* may be driven by the higher abundance of frequently occurring rather than rare phenotypes. However, even though the estimated number of undetected phenotypes was higher for *P. oceanica* compared to *P. crispa*, the overall estimated diversity for all orders of $q$ in the red algae

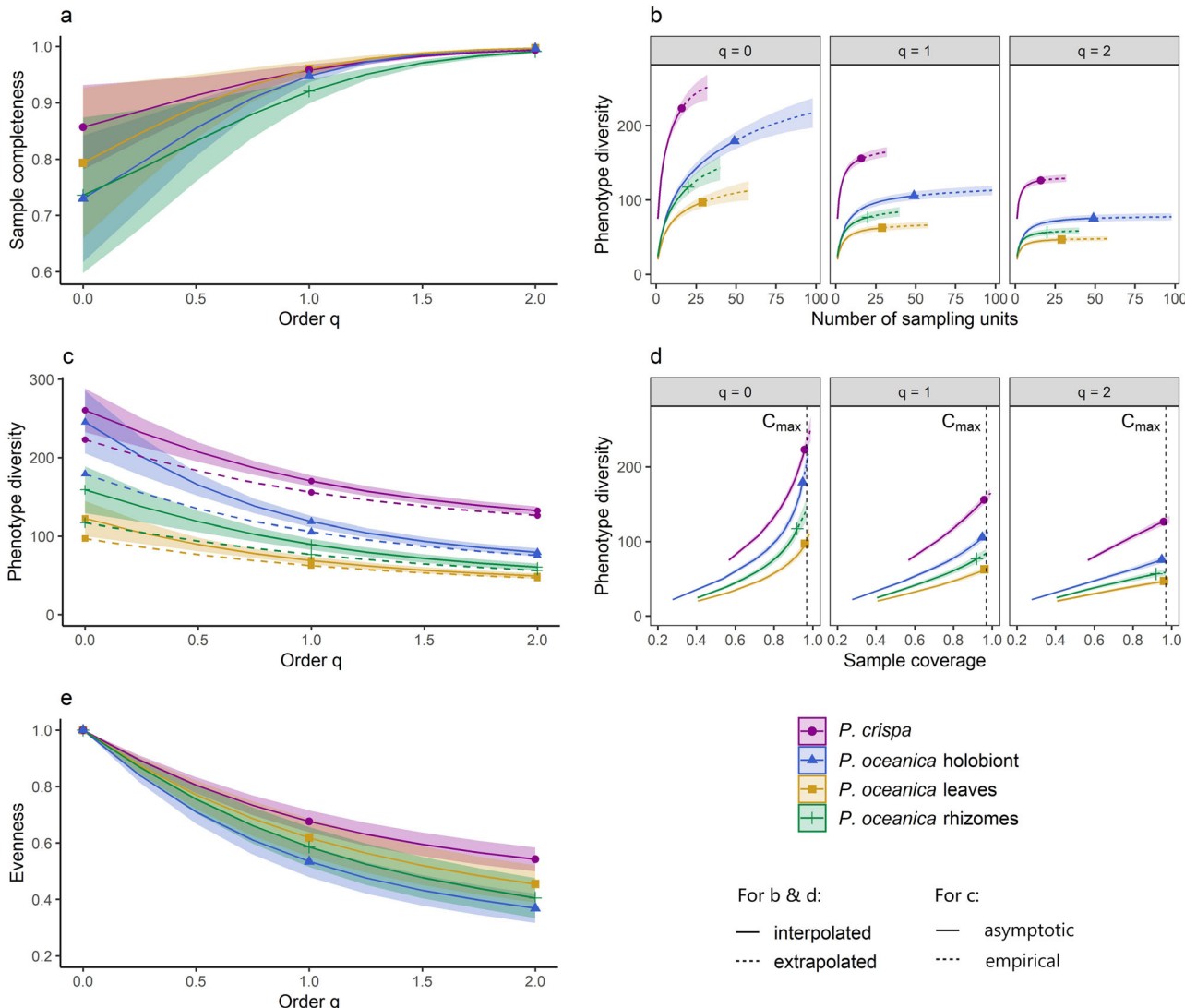

**Fig. 3 Overview of biodiversity analysis based on Hill numbers. a** Estimated sample completeness curves as a function of order $q$ between 0 and 2. **b** Size-based rarefaction (solid lines) and extrapolation (dashed lines) curves up to double the respective sample size. **c** Asymptotic estimates of diversity profiles (solid lines) and empirical diversity profiles (dashed lines). **d** Coverage-based rarefaction (solid lines) and extrapolation (dashed lines) curves up to double the reference sample size. Vertical dashed lines show the standardised sample coverage $C_{max} = 96.6\%$. **e** Evenness profiles as a function of order $q$, $0 < q \leq 2$, based on the normalised slope of Hill numbers. Dots (*P. crispa*), triangles (*P. oceanica* holobiont), rectangles (*P. oceanica* leaves) and crosses (*P. oceanica* rhizomes) denote observed data points. All shaded areas in **a**–**e** denote 95% confidence intervals obtained from a bootstrap method with 500 replications. Note: some bands are invisible due to narrow width.

habitats still remained higher relative to the seagrass meadows (Fig. 2c). Taken together, our data have identified *P. crispa* as a habitat that harbours more even and diverse sessile invertebrate communities compared to neighbouring *P. oceanica* meadows.

**Red algae mats fulfil ecosystem engineer functions.** We measured key environmental parameters (i.e., oxygen concentrations, light availability, pH, temperature, chlorophyll α concentration, and water movement) in neighbouring *P. crispa* and *P. oceanica* to assess *P. crispa*'s functioning as an ecosystem engineer. Our results suggest that *P. crispa* shapes key environmental parameters similarly to neighbouring *P. oceanica* seagrass meadows (Fig. 4). In particular, water movement and light intensity within the red algae mats and in the seagrass meadows were lower than for the neighbouring bare substrate (Fig. 4b, f). This extends the findings of a parallel study that has identified *P. crispa* as an ecosystem engineer modifying its environment[33]. This functioning as an ecosystem engineer seems to

apply to further environmental parameters: daily oxygen concentration fluctuations of *P. crispa* (7.73–8.14 mg l$^{-1}$) were similar to those of *P. oceanica* (7.59–8.04 mg l$^{-1}$), with the daily mean of oxygen concentrations being slightly higher in *P. crispa* (7.99 mg l$^{-1}$) compared to those of *P. oceanica* (7.75 mg l$^{-1}$). This contradicts previous findings stating that shallow, macroalgae-covered environments undergo wider oxygen concentration fluctuations compared to seagrass meadows[53,54]. Our findings indicate that this may not necessarily be the case in deeper environments (Fig. 4a). In addition, the average pH within *P. crispa* mats was lower (8.44) compared to *P. oceanica* meadows (8.64), which resembled the observed differences in O$_2$ concentrations (Fig. 4a, c). Photosynthesis by algae and plants requires hydrogen ions, which results in increased pH levels while respiration lowers pH levels[55,56]. Furthermore, our data suggest higher light availability in *P. crispa* (538 lux) compared to *P. oceanica* meadows (315 lux; Fig. 4b) at the same depth. These findings corroborate with previous studies that identified strong light attenuations in seagrass macrophyte habitats due

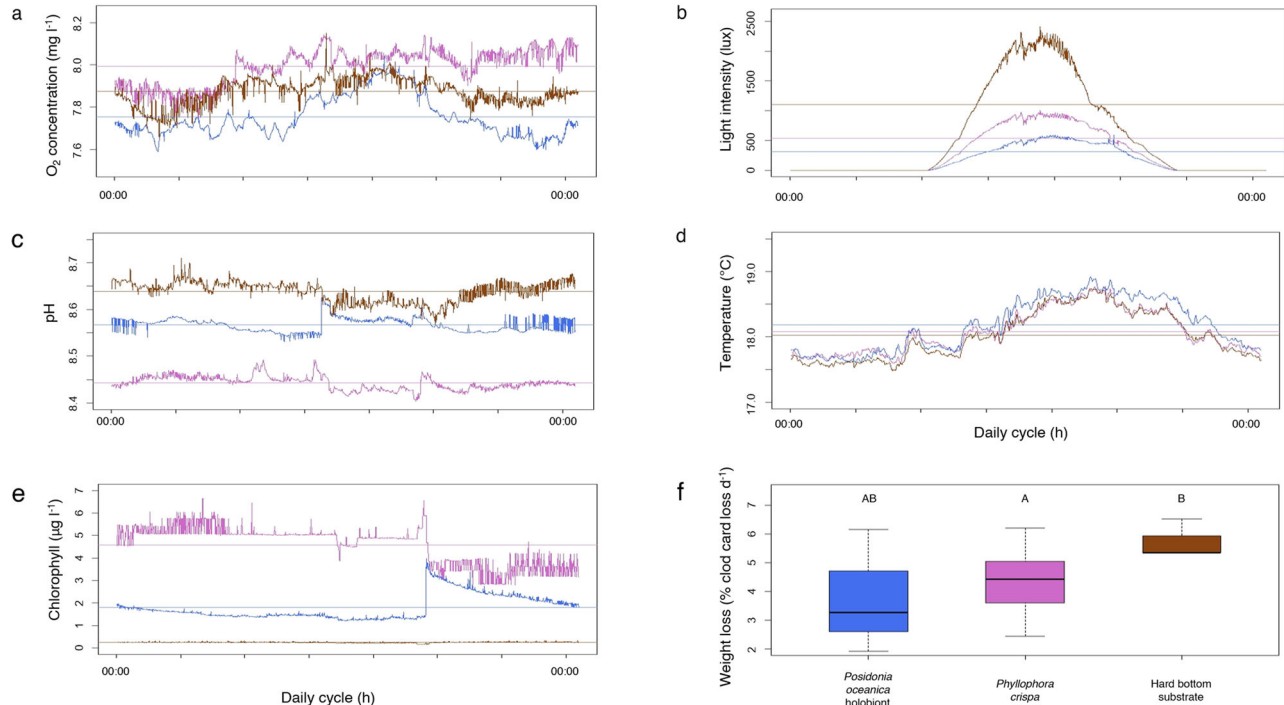

**Fig. 4 Environmental parameters measured in *Phyllophora crispa* and *Posidonia oceanica*.** Environmental data consisting of oxygen ($O_2$) concentration (**a**), light intensity (**b**), pH (**c**), temperature (**d**), chlorophyll *a* concentration (**e**) and water movement (estimated via weight loss of clod cards; **f**) in *Phyllophora crispa* (purple), *Posidonia oceanica* (blue) and neighbouring hard-bottom substrate serving as a reference habitat (brown). Horizontal lines within panels **a**–**e** display daily mean of respective deployment (with $n = 13$ for *P. crispa*, $n = 7$ for *P. oceanica*, $n = 6$ for reference habitat for $O_2$ concentration, pH and chlorophyll *a* concentration, and with $n = 10$ for *P. crispa*, $n = 6$ in *P. oceanica*, $n = 4$ in reference habitat for light intensity and temperature) of respective parameters in each habitat. Note for panel **f**: different letters above box plots indicate significant differences between habitats (ANOVA and subsequent Tukey HSD test), with $n = 9$ for *P. crispa*, $n = 4$ for *P. oceanica*, $n = 2$ for reference habitat.

to self-shading effects[57,58]. A lessened self-shading effect in the red algae habitat compared to the *P. oceanica* seagrass meadows could be explained by morphological differences between the two habitats. The latter forms meadows of higher thickness relative to the mats formed by *P. crispa*, with *P. oceanica* leaves being wider than thalli of *P. crispa*. Finally, the reduced water movement (Fig. 4f) in both habitats and higher $O_2$ availability in *P. crispa* compared to *P. oceanica* (Fig. 4a) may benefit the settlement of specific bryozoans (e.g., *Bugula* sp., *Schizoporella* sp.)[59–63], bivalves[64] and polychaetes (*Hydroides* sp.)[63]. This may explain the findings of the present study, in which we identified moror individuals associat/or individuals associated with *P. crispa* compared to *P. oceanica* of bryozoans (76 vs. 78 phenotypes, 44,222 vs. 7655 ind habitat m$^{-2}$), molluscs (bivalves; 4 vs. 4 phenotypes, 112 vs. 38 ind habitat m$^{-2}$) and polychaetes (23 vs. 13 phenotypes, 5950 vs. 3734 ind habitat m$^{-2}$[65]; see Supplementary Table S1). Potentially, lower pH in *P. crispa* mats (Fig. 4c) may have limited the presence of organisms such as bivalves[66] or benefitted comparatively resilient organisms such as specific bryozoans[67]. Hence, the extent to which lower pH conditions in *P. crispa* compared to *P. oceanica* may have counteracted potential benefits such as higher $O_2$ availability (Fig. 4a) remains speculative.

The higher number of phenotypes and individuals in *P. crispa* relative to *P. oceanica* may be partly explained by the specific surface area that potentially offers substrate, and thus micro-habitats for mobile and sessile invertebrates[27,30]. The complex morphology of *P. crispa* mats is reflected in the 2D to 3D surface area enlargement factor. Here, a high surface area provided by complex thalli relative to a small volume (mats of several cm thickness) resulted in an enlargement factor of $4.9 \pm 0.2$ (mean ± standard error; Supplementary Table S5) for *P. crispa*,

which was lower than for *P. oceanica* (both leaves ($7.3 \pm 0.5$) and the *P. oceanica* holobiont ($8.3 \pm 0.5$) but higher than for *P. oceanica* rhizomes ($2.0 \pm 0.1$)). This structural complexity may also explain the observed reduced water movement within *P. crispa* mats (Fig. 4f) that could favour sediment trapping. The extent to which further functions such as sediment trapping, similar to the reduced water movements induced by *P. oceanica* meadows[68–70], apply to *P. crispa* mats needs to be determined in future studies. Trapped sediment and particulate matter could provide (1) a heterogeneous habitat for infaunal species[71] and (2) (in-) organic matter for tube-building species such as sessile polychaetes[71]. Growth form, enlargement factor and persistence[30] of *P. crispa* contradict the common notion that structural complexity is reduced when spatially complex and long-living habitats, such as seagrass meadows, decline[2,6]. We further estimated the number of individuals per area m$^2$ of seafloor by multiplying the calculated numbers of individuals per habitat m$^2$ with the respective enlargement factor (Supplementary Table S5). *P. crispa* supported $313,635 \pm 27,486$ ind seafloor m$^{-2}$, which was approximately twice that of the *P. oceanica* holobiont ($162,139 \pm 11,794$ ind seafloor m$^{-2}$; Dunn's test $p < 0.001$; Supplementary Table S4).

We conclude that *P. crispa* mats facilitate the colonisation of sessile organisms[27,72] by providing (micro-) habitats for associated alpha diversity (Table 1 and Fig. 3), thus, allowing us to propose *P. crispa* as an ecosystem engineer[1,73]. Together with the considerable surface area enlargement (Supplementary Table S5), environmental parameters shaped by *P. crispa* (Fig. 4), its wide distribution[27,29,31,32] and the comparative biodiversity analysis (Table 1 and Figs. 2 and 3), red algae mats may function as overlooked ecosystem engineers and harbour high sessile invertebrate biodiversity.

**Fleshy red algae as refuge habitat**. Like many other marine ecosystems, *P. oceanica* seagrass meadows experience a range of anthropogenic threats, which have caused a drastic decline in the spatial distribution throughout the Mediterranean[6]. The loss of biodiversity is only one among many consequences of declining *P. oceanica* meadows[5,6,19]. The high biodiversity associated with red algae *P. crispa* mats may positively impact sessile invertebrate communities in bordering *P. oceanica* seagrass meadows[74], which is reflected by a total of 90 shared phenotypes that occurred in all investigated habitats (Fig. 2a).

Even though *P. crispa* mats harboured sessile invertebrates in numbers that exceeded those of neighbouring *P. oceanica* meadows[65] and other ecosystems (Table 1), these mats substantially differed from seagrass meadows in their longevity. In the Mediterranean, *P. oceanica* meadows form dense rhizome layers that can be of several metres of thickness when admixed with trapped sediment[24]. Similar to coral reefs or mangrove forests, seagrass meadows can persist for several millennia[75], which exceeds the currently estimated lifespan of *P. crispa* formations (i.e., decades)[30]. The evolved size and physical structure of seagrass meadows can result in a dissipation of wave energy on multiple levels (reviewed in ref. [23]) and reduce coastal damage and erosion. Wave energy is a key limiting factor defining the upper physical boundary that shapes the bathymetric spatial distribution for *P. oceanica* meadows[76]. The properties of *P. oceanica* allow it to withstand these physical impacts and grow at depths as shallow as 0.5 m[77]. In contrast to *P. oceanica* meadows, *P. crispa* mats can be dislodged and translocated by waves[30], particularly those with an unattached growth form on sediments[31]. Although dislodged *P. crispa* may not offer a stable environment over longer time scales, mobile algal thalli may function as an effective dispersal mechanism. Drifting algae parts may offer substrate to diverse sessile invertebrate communities[35,36] and function as a transport vector over large distances[37]. The extent to which the associated phenotypes identified in this study tolerated this drifting behaviour remains speculative[38]. The translocation of *P. crispa* mats may have consequences for associated biodiversity through two pathways: (i) translocated *P. crispa*[30], which can colonise and spread vegetatively, may still provide habitat for associated sessile invertebrates; or (ii) *P. crispa* mats are severely damaged, losing their function as ecosystem engineers, and, hence, biodiversity hotspots. We conclude that in both cases, *P. crispa* mats serve as *temporary* ecosystem engineers forming *temporary* refuge habitats, and subsequently as *transitory* biodiversity hotspots. Potentially, more tolerant sessile species could reach more favourable areas such as healthy seagrass beds that are possibly beyond the reach of planktonic larval stages. *P. crispa* formations in the Atlantic and Black Sea provide a relatively stable habitat over several decades[30], which underlines the general functioning as a biodiversity substratum. The extent to which this function applies to *P. crispa* mats of the Mediterranean as well needs to be determined in future studies.

We postulate that sessile invertebrates can re-colonise recovering *P. oceanica* meadows, if appropriate conservation measures are implemented[18,78]. Seagrass meadows can recover from anthropogenic or natural threats on a decadal timescale[79], which corresponds with the lifespans of *P. crispa* mats[30]. Hence, red algae mats may function as overlooked biodiversity refuge habitats supporting the recovery of classical habitats such as seagrass meadows, particularly due to their proliferation across the Mediterranean[27–29], the Black Sea[30,31] and the Atlantic[29,32]. Likewise, similar patterns (i.e., the supported recovery of a habitat by neighbouring habitats) were reported from the Great Barrier Reef, where the recovery of a bleached reef was facilitated by larval inflows originating from non-bleached reefs[80]. In the Mediterranean, we hypothesise that *P. crispa* can support *P. oceanica* meadows (and other habitats, see Table 1) by maintaining their sessile invertebrate biodiversity[74,81], particularly due to an overlap of shared phenotypes, i.e., sessile invertebrates that occurred in both *P. crispa* and *P. oceanica* (Fig. 2a), even though both habitats harbour a range of unique phenotypes. It remains to be determined (i) to what extent the community composition in re-colonised *P. oceanica* meadows differs from their initial sessile invertebrate community composition, considering the clear distinction of associated sessile invertebrate communities in *P. crispa* mats and *P. oceanica* meadows (Fig. 2b), and (ii) whether this function applies to all shared phenotypes and potentially further taxa. Our findings suggest that *P. crispa* mats and their associated sessile invertebrate communities potentially aid in reviving classical marine (sessile invertebrate) biodiversity hotspots such as invaluable seagrass meadows in the Mediterranean Sea once threats are reduced or removed[17,79,82].

## Methods

**Study site and sampling**. All data were generated by SCUBA diving between May and July 2019 along the north-eastern and north-western coasts of Giglio Island, within the Tuscan Archipelago National Park, Tyrrhenian Sea, Italy (Supplementary Fig. S1). Samples for biodiversity assessments were taken at six sites (two each for *P. crispa* mats of >5 cm thickness and *P. oceanica*, and two for co-occurring habitats, resulting in four sampling sites for *P. crispa* and *P. oceanica* each, see Supplementary Fig. S1) according to accessibility and occurrence of target habitats at water depths between 28 and 30 m.

To sample *P. crispa* mats for the present study, a sampling frame (30 × 30 cm) was randomly placed in the target area four times (i.e., each time 50 cm apart), and all algal material within the frame was carefully removed using a spatula and subsequently placed into 1 L PP-bottles (each holding a ratio of algae:water = 1:3). A total of 16 replicates for *P. crispa* were sampled. *P. oceanica* rhizome and leaf specimens were sampled separately into 1 L Kautex jars to avoid oxygen depletion or physical damage during transport. An attached growth form of *P. crispa* was chosen for the present study. *P. oceanica* root-rhizomes were cut including the sheaths, both vertical and horizontal rhizome as well as the upper layers of the roots (Supplementary Fig. S2, hereafter referred to as '*P. oceanica* rhizome'). Leaves were cut with scissors directly at the sheath of the shoot. A total of 20 *P. oceanica* rhizome specimens and 29 single leaves were collected from the four sampling sites to minimise the impact on threatened *P. oceanica* meadows. The number of sampled specimens at the respective sampling locations was 4× 'Corvo', 4× 'Fenaio', 4× 'Punta del Morto', and 4× 'Secca 2' for *P. crispa*; 10× '3 Fratelli', 5× 'Fenaio', 5× 'Cala Calbugina', and 9× 'Secca 2' for *P. oceanica* leaves; 10× '3 Fratelli' and 10× 'Secca 2' for *P. oceanica* rhizomes (see Supplementary Fig. S1).

All samples (*P. crispa*, *P. oceanica* leaves and rhizomes) were transferred immediately to the seawater husbandry tanks of the Institute for Marine Biology (IfMB, located on the island of Giglio, Italy) upon return from sea under stable physical conditions (18 °C, 12:12 h dark/light cycle, light similar to in situ conditions) until further analysis. For biodiversity assessments, four subsamples of *P. crispa* were taken from each of these main samples.

**Biodiversity assessment**. All samples were analysed within three days after collection. For *P. crispa*, subsamples (sensu Bianchi (2004)[83]) were transferred to plastic bowls, where *P. crispa* mats were cut into single thalli, and subsequently placed in single Petri dishes. Thalli were then analysed using stereo magnifiers (max. ×40 magnification) to determine invertebrates that were assigned to one of the following taxa: Ascidiacea, Bryozoa, Cnidaria, Entoprocta, Foraminifera, Mollusca (Bivalvia), Polychaeta (Sedentaria), Rotifera, and Porifera. Foraminifera were determined using a microscope (max. 400x magnification). Seagrasses such as *P. oceanica* are typically divided into two sub-habitats: the leaf canopy-forming part and a dense root-rhizome layer[48,84], both varying in their habitat characteristics and associated biotic assemblages[85,86]. Thus, we investigated the sessile invertebrate diversity in both sub-habitats in our analysis by assessing invertebrate phenotype abundances separately for *P. oceanica* leaves and rhizomes to account for potential differences. *P. oceanica* rhizomes were analysed as a whole using a stereo microscope, whereas *P. oceanica* leaves were cut into pieces of ~8 cm length for handling and to avoid double counting. All *P. oceanica* samples were analysed for the aforementioned taxa as well. All specimens were identified according to relevant literature (Supplementary Table S7) and crosschecked online with the World Register of Marine Species (marinespecies.org). Individual specimens or colonies in case of colonial species (i.e., Bryozoa) were then counted for further analysis. In case no clear identification was possible, individuals were distinguished based on distinct visual characteristics, resulting in a dataset consisting of distinct phenotypes rather than species. We refer to Supplementary Table S8, which consists of a subset exemplarily showing the applicability via a clear correlation of the number of species and phenotypes, respectively. Finally, all numbers were normalised to their respective habitats' surface area using the corresponding enlargement factor (see next section), resulting in a total number of individuals per habitat m².

To test for statistical differences between the number of individuals among habitats, a Shapiro–Wilk test for normality, Kruskal-Wallis-test and a subsequent post-hoc Dunn's test were performed in R (version 4.0.4)[87] with the interface RStudio (version 1.0.153)[88] using the 'shapiro.test', 'kruskal.test' and 'dunnTest' functions from the 'stats'[87] and 'FSA'[89] packages. We expected numbers in *P. oceanica* leaves and rhizomes to exceed those of *P. crispa* given higher sampling efforts for the former ($n = 29$ and $n = 20$, respectively vs. $n = 16$). To allow for comparisons among habitats—despite differences in sampling efforts—we applied a combination of asymptotic and non-asymptotic diversity estimations based on rarefaction and extrapolation analysis tools, and Hill numbers (see below). We used phenotype incidence instead of abundance data, as diversity estimations based on Hill numbers rely on species (or phenotypes in the present study) occurring as singletons (i.e., occurring in one sample or with abundances of one individual). Given that we normalised phenotype abundance counts to habitat and seafloor area (m²) to enable comparison between habitats, the assemblages sampled by us are devoid of singleton occurrences, ultimately leading to samples appearing complete in terms of capturing true diversity. This is highly unlikely with a non-exhaustive sampling effort and we, thus, opted to use phenotype incidence data for diversity and sample completeness estimation, as this has been shown to not be statistically inferior for the use of count abundances (e.g., ref. [90] and ref. [49]).

A statistical biodiversity assessment was performed using a combination of the *iNext4steps* online tool (https://chao.shinyapps.io/iNEXT4steps/) and the 'iNext' package[91] in R (version 4.0.4)[87] with the interface RStudio (version 1.0.153)[88]. Given that the official online tool was not yet available at this time, Chao et al.[51] provided a hyperlink to a trial version that we used in this study. Plots were created using 'iNext's ggiNext' function and the 'ggplot2' package[92]. We refer to Daraghmeh and El-Khaled[93] for a detailed workflow and scripts. Briefly, to assess and compare sample completeness and alpha diversity of the respective habitats, we followed the protocol proposed by Chao et al.[51]. It is based on their extensive earlier works (e.g., Chao et al.)[49], which use the now widely accepted concept of Hill numbers, also known as the effective number of equally abundant species $^qD$[49,50]. Here, $q$ denotes the diversity order of a Hill number and determines its sensitivity to species' relative abundances or frequencies (in case of incidence data, i.e., species presence/absence). Hill numbers based on higher values of $q$ put more emphasis on more commonly occurring species. The most widely used members of the family of Hill numbers are the ones of orders $q = 0$, $q = 1$ and $q = 2$. For sampling-unit-based phenotype incidence data as used in the present analysis (see below), $^0D$ indicates the measure of phenotype richness (i.e., all phenotypes are quantified equally without regard to their actual frequencies) and $^1D$ and $^2D$ represent Shannon (i.e., exponential of Shannon entropy) and Simpson (i.e., inverse of Simpson concentration index) diversity, i.e., the effective number of frequent and highly frequent phenotypes, respectively[51]. Here, we used phenotype incidence data as described above.

The calculation of Hill numbers based on sample data (i.e., empirical or observed Hill numbers) is biased regarding sample completeness and size[49]. We followed the workflow and steps listed below to achieve meaningful comparisons of the investigated biotic communities (see ref. [51] and Supplementary Table S6):

(I) Estimation of sample-completeness profiles from sample data via a bootstrap method ($n = 500$) to obtain confidence intervals: this enabled comparison of sample completeness (i.e., diversity detected) of our various habitat datasets. Profiles that increase with order $q$ indicate incomplete sampling and therefore undetected diversity.

(II) Empirical and asymptotic estimation of true diversities based on hypothetical large sample sizes[94]: sufficient data are a prerequisite for the latter, however. To investigate if our data fulfilled this requirement, we computed sample-size-based rarefaction and extrapolation (R/E) sampling curves for Hill numbers of different orders[49,90]. Extrapolation was performed to double the actual number of samples per habitat, as further extrapolation is unreliable in the case of phenotype richness[90]. Levelling out of R/E curves indicates that asymptotic estimates are accurately representing true diversities. In this case, asymptotic and empirical Hill numbers may be compared to assess undetected diversity and the comparison of asymptotic diversity profiles allows the assessment of differences in diversity between habitats. If R/E curves do not level out, asymptotic diversity estimates represent true diversity only up to a certain level of sample coverage (i.e., $C_{max}$, see below) and, therefore, have to be considered as minimum estimates of true diversity.

(III) Comparing diversity for a non-asymptotically standardised sample coverage (i.e., sample completeness for $q = 1$) in the case where asymptotic estimation of true diversity is unreliable: diversity may then be compared between equally complete samples. Here, coverage-based R/E curves were computed to the maximum coverage $C_{max}$. This value represents the sample coverage of the habitat exhibiting the lowest coverage when samples are extrapolated to double the respective number of samples per habitat.

(IV) Estimation of evenness profiles for $q > 0$ at $C_{max}$ based on ref. [95]: to compare evenness profiles of assemblages with varying levels of richness, the slopes of Hill-number diversity profiles connecting two points at $q = 0$ and any $q > 0$ are being analysed, whereby steeper slopes represent higher unevenness of phenotype incidences. The slopes were normalised and converted to an evenness value. This was possible for orders of $q > 0$, but not for $q = 0$, as all phenotypes are accounted for equally in the latter. In addition, Pielou's $J'$ was calculated as a phenotype evenness measure based on Hill numbers of $q = 0$ and 1[95,96]. Both evenness

profiles and Pielou's $J'$ are based on the richness and Hill-number diversity and were therefore estimated at a standard level of $C_{max}$.

**Biodiversity indices for study comparison and community composition analysis.** Due to missing original data of studies investigating sessile invertebrate biodiversity hotspots in the Mediterranean and elsewhere (see Table 1), but to ensure comparability with the present study, classical alpha biodiversity (Shannon, Simpson) indices, as well as Evenness index not based on Hill numbers were calculated as followed[50]:

$$\text{Shannon index} - \sum_i \left( \frac{n_i}{N} \cdot \log_2 \left( \frac{n_i}{N} \right) \right) \quad (1)$$

$$\text{Simpson index} \frac{\sum_i n_i (n_i - 1)}{N(N-1)} \quad (2)$$

$$\text{Evenness index} - \frac{\sum_i \left( \frac{n_i}{N} \cdot \ln \left( \frac{n_i}{N} \right) \right)}{\ln N} \quad (3)$$

where $n_i$ is the number of phenotypes/species in a taxon, and $N$ is the total number of taxa, with a maximum of 9 as previously defined.

Non-parametric permutational multivariate analysis of variance (PERMANOVA[97]; based on species abundance data using Primer-E v6[98] with the PERMANOVA+ extension)[99] was used to check for significant differences (i.e., $p \leq 0.05$) in the sessile invertebrate community composition among (sub-) habitats. For this, raw count data (related to habitat m²) were square-root transformed to generate Bray–Curtis similarity matrices for PERMANOVAs with habitats as a factor. Pair-wise PERMANOVA tests were then performed with the unrestricted permutation of raw data (999 permutations), Type III (partial) sum of squares and Monte Carlo tests. In case pair-wise comparisons exhibited significant differences, we checked if these differences may partially or fully be driven by the heterogeneity of multivariate dispersion. In addition, differences in the sessile invertebrate community composition were visualised by applying nMDS based on Bray–Curtis similarities. To exclude the parameter 'sampling location' as a major driver shaping biodiversity patterns, an nMDS plot based on Bray–Curtis similarities was performed (see Supplementary Fig. S3). A PERMANOVA was performed based on the similarity calculations and on Bray–Curtis similarities (incidence data), in order to test for differences between (sub-) habitats. Lastly, an area-proportional Venn diagram was constructed to describe the shared and unique phenotypes among (sub-) habitats, i.e., *P. crispa* mats, *P. oceanica* leaves and *P. oceanica* rhizomes.

**Surface area quantification.** For *P. crispa*, the wet weight of sub- and main samples (see above) was measured after taking algal material of approximately 10 g and shaking off excess water three to five times with one hand. The subsamples were then placed in a bowl on a laminated grid paper and flattened with a glass pane, ensuring that thalli parts did not overlap. Then, pictures were taken from above at a 90° angle using a Canon G12 digital camera and a monopod stand (KAISER RS1) to ensure a constant distance and angle to the respective thallus. The surface area of the algae in the subsamples was then calculated from the picture using ImageJ (version 1.52)[100] and multiplied by two to consider both sides of the thalli. The surface area and enlargement factor of the main sample were then calculated as followed:

$$SA_M = \frac{WW_{MS} * SA_{SS}}{WW_{SS}}$$

$$EF_{PC} = \frac{SA_{MS} + 0.09\, m^2}{0.09\, m^2}$$

where $WW$ is the wet weight, $SA$ the surface area, $MS$ the main sample, $SS$ the subsample, and $EF$ is the enlargement factor (i.e., 0.09 m² corresponds to the area of the sampling frame).

For *P. oceanica*, the commonly applied Leaf Area Index[101] was extended to include *P. oceanica* rhizomes in the surface area calculation. The surface area of *P. oceanica* was modelled using advanced geometry (sensu ref. [102]), as a cylindrical shape was assumed for the rhizomes and a rectangular shape for the leaves. Both the length and width of the leaves were measured with a ruler. Subsequently, the number of leaves was determined at the sheath of each rhizome. During additional sampling dives, rhizome density was counted 16 times using a 40 × 40 (=0.16 m²) sampling frame. Following this, the enlargement factor was calculated according to:

$$EF_{PO} = \frac{[SA_{rhizome} + (SA_{leaves} \times leaves/rhizome)] \times rhizome/m^2 + 0.16\, m^2}{0.16\, m^2}$$

For reference purposes, the surface area enlargement factor of neighbouring bare granite/hard-bottom substrates was calculated as well using a 20 cm × 20 cm × 2.9 cm PVC-frame (RA = 0.04 m²) with ball chains (metal ball diameter = 2.4 mm) attached to at least three of the four corners of the frame. The chains served to trace the actual dimensions (diagonals and edges) of the underlying substrate enclosed by the projection of the frame's planar dimensions onto the sample surface. Metal chains were laid out from corner to corner of the frame whilst being aligned to the uneven sample surface. The ball chain link numbers up to the intersection point with the corners of the frame were counted

and converted into the equivalent distance. Using these values, an estimation of the actual surface area could be calculated using Heron's formula[103] (Supplementary Method 1). This was done by calculating two triangular partial surfaces for one diagonal each. The surface area of the underlying substrate was calculated twice (1× for each diagonal), to generate a mean value as an estimate of the actual surface area. A total of 15 frames were sampled to form a mean value for the inorganic surface magnification factor of the granite substrate.

**Environmental parameters**. Environmental parameters were assessed in situ at a depth of 28–30 m close to the Punta del Morto dive site (42°23'22.2"N 10°53'24.3"E; Supplementary Fig. S1) of Giglio Island in September and October 2019, where all target habitats (i.e., *P. crispa* mats of >5 cm thickness, *P. oceanica* seagrass meadows of >20 cm height, hard-bottom substrate serving as a reference habitat for environmental parameter assessments) were found less than 10 m apart from each other. Thus, all habitats likely experienced similar environmental conditions allowing a direct comparison of environmental parameters between the habitats. Oxygen concentration, pH, and in situ Chlorophyll (Chl) α-like fluorescence were obtained from Eureka Manta logger (GEO Scientific Ltd.) that recorded data at 1-min intervals. Chl α-like fluorescence was measured with an optical sensor with a light-emitting diode at an excitation wavelength of 460 nm and emission wavelength of 685 nm (resolution of $0.01\,\mu gL^{-1}$ and accuracy of ±3%). Manta loggers were deployed multiple times (13× in *P. crispa* mats, 7× in *P. oceanica* meadows, 6× in reference habitat) for 2–3 days.

Water movement within the habitats was measured using clod cards[104,105]. Gypsum (Quick-mix gips, toom #3050388 $CaSO_4$) clod cards (hereafter GCC) were produced and constant dry-weighted before deployment. The GCCs were placed 1 cm above the seafloor (i.e., within the mat) and 20 cm above the mat. They were attached to a metal stick with a 90° offset (Supplementary Fig. S4) to address water movement within and above *P. crispa* mats. An identical GCC setup was used on hard-bottom substrates as a reference. Due to differences in height, the setup was adjusted for *P. oceanica* meadows, i.e., GCCs were placed 20 cm above the seafloor within the meadows, as well as 20 cm above the meadows. All setups were assembled prior to deployment and were positioned multiple times (9× in *P. crispa* mats, 4× in *P. oceanica* meadows, 2× in reference habitat) for 6–7 days in the respective habitat. Afterwards, GCCs were cautiously transported, rinsed with freshwater, and dried at 60° until they reached a final constant weight. The difference in weight prior to and post deployment was related to deployment time, resulting in weight loss $d^{-1}$ as an indicator for the strength of the relative water movement. This allowed for relative comparisons within and among *P. crispa*, *P. oceanica* and hard-bottom substrates. To check for statistically significant differences in the water movement in *P. crispa* compared to *P. oceanica* and hard-bottom substrate, an analysis of variance and subsequent post-hoc test (Tukey HSD) was performed.

Similar setups were prepared for light intensity and temperature assessments. Instead of GCC, multiple Onset HOBO Pendant Data Loggers (part #UA-002-64) were placed (10× in *P. crispa* mats, 6× in *P. oceanica* meadows, 4× in reference habitat) accordingly, recording data at 15-sec intervals for 5 consecutive days. A total of 10–18 days and respective data points covering every 15 s (in case of Onset HOBO Pendant Data Loggers) or every minute (in case of Eureka Manta Loggers) of a daily cycle were collected.

**Reporting summary**. Further information on research design is available in the Nature Research Reporting Summary linked to this article.

## Data availability
All data are freely available from the corresponding author and accessible via El-Khaled et al. (2021)[52].

## Code availability
All code is available from the corresponding author and accessible via Daraghmeh and El-Khaled (2021)[93].

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

## Acknowledgements

We are thankful to Dr Jenny Tuček as well as Rainer and Regina Krumbach (Campese Diving Center) and Dr med. dent. Ulrich Merkel for logistical support. We are also thankful to Mischa Schwarzmeier and Annette Reh for support during our fieldwork, as well as Alice G. Bianchi, Beltrán Montt and Dr Susann Roßbach for support during sampling activities. The authors are thankful to Claudia E. L. Hill for her thorough style edits on the manuscript.

## Author contributions

C.W. and Y.C.E.K. conceptualised and designed research. F.I.R., M.B., R.M., J.P., N.S., L.W., B.M., C.W. and Y.C.E.K. performed research. Y.C.E.K. and N.D. coordinated data analyses with contributions of A.T., A.K. and F.I.R.. Y.C.E.K. and C.W. wrote original draft of the manuscript with significant contributions of N.D., A.T., F.R., M.H., F.I.R., E.C., A.K., M.B., R.M., J.P., N.S., L.W. and B.M. All authors read and approved the final manuscript.

## Funding

## Competing interests

The authors declare no competing interests.
