## [Peer Review File · Communications Biology]

Reviewers' comments:

Reviewer #1 (Remarks to the Author):

COMMSBIO-21-2198

Fleshy red algae mats act as reservoir for sessile invertebrate biodiversity

This manuscript presents the results of a comparison of sessile invertebrate biodiversity in persistent fleshy red algae mats and *Posidonia oceanica* seagrass beds in the Mediterranean. The authors show that sessile invertebrate biodiversity is higher in the former than the latter and suggest that these algae mats may act as biodiversity reservoirs when *P. oceanica* beds are in decline or recovery. I generally agree that this type of study is important to publish, because it can provide nuance to and/or challenge paradigms that inform environmental policy decisions. However, I feel the manuscript would benefit from addressing some important points about the design, analysis, applicability, and overall message of the study.

General comments:

1. I think the manuscript does not emphasize enough that the study pertains to sessile invertebrate biodiversity. This is, of course, directly stated in the title, but it is glossed over in other parts of the manuscript. The result is that the manuscript both oversells the findings (by appearing to extend the findings to biodiversity too broadly) and undersells the findings (by not including much discussion of why we should care about sessile invertebrate epifauna and their effects on ecological functioning in these macroalgal and seagrass habitats, relative to other groups that have received more attention). I think adjustments are advisable throughout the manuscript – some specific suggestions are below – but are especially important in the abstract.
2. It would be helpful for the introduction to justify why sessile invertebrate epifauna were specifically targeted for this study. p. 4 L. 75 suggests that *P. crispata* is a general diversity hotspot, but the cited study is specific to serpulid polychaetes. Seagrass beds are known for hosting diverse assemblages of many functional groups: not only sessile invertebrate epifauna, but also epiphytes, mobile invertebrate epifauna, fish, meiofauna, and macrofauna. Among these groups, why were sessile invertebrate epifauna chosen for special attention here? Was sampling for mobile invertebrate epifauna considered? Mobile invertebrate epifauna have been included in numerous past comparisons of seagrasses and macroalgae, including the Thomsen et al. 2012 study cited in this manuscript to justify potentially considering *P. crispata* a reservoir.
3. The authors' argument that red algae mats fulfill ecosystem engineering functions could be strengthened by providing further justification for the methods, and by putting the results more fully in context of the other ecosystem engineers to which the algae are being compared (here, seagrasses, specifically *P. oceanica*) and the organisms benefitting from their modification of the environment (here, sessile invertebrates). First, is there additional support for the idea that 5 days of environmental data from a single site are representative of most or all of the other sites examined? Second, can more detail about how similarity between habitats was assessed be provided? Dissolved oxygen and pH, for example, may differ between seagrass beds and macroalgae mats, but the difference is often in degree of fluctuation rather than level per se (e.g. Viaroli et al. 2008), and this may or may not be obvious in simple plots of the data. At the very least, perhaps the cited findings in Schmidt et al. 2021 could be elaborated upon. Third, seagrasses affect their environment in ways other than the parameters listed, such as stabilizing sediment and profoundly influencing geochemical fluxes in the water column and sediment. Not mentioning any of these makes sense only if the focus is clearly on sessile invertebrate epifauna throughout the manuscript.
4. More context for the authors' suggestion that *P. crispata* mats may act as reservoirs of sessile invertebrate biodiversity would be helpful. Are there groups of sessile invertebrates in *P. crispata* mats that the authors feel would particularly assist in re-establishing *P. oceanica* beds or their functions (e.g. filter feeders that help to maintain water clarity)? A discussion earlier in the results of which sessile invertebrate phenotypes or taxa are responsible for distinguishing the three habitats examined would also be useful – I think the argument that *P. crispata* mats can act as

reservoirs for *P. oceanica* because their community taxa overlap so much is blunted in the text and Figure 2 by the very clear distinction in community compositions from the nMDS analysis.

5. I would urge the authors to carefully consider the clarity and flow of their writing during the revision process, especially in the Methods section (and in consideration of the fact that the Results and Discussion section will be read before the Methods section).

Specific comments:

6. p. 2 L. 29-32: I recommend including the words sessile and/or sessile invertebrate somewhere in these concluding sentences (e.g. "alternative habitats and sessile invertebrate biodiversity reservoirs..."). If the reader misses the "sessile" on line 28, the last two sentences of the abstract imply that these red algae mats are habitats of equivalent general biodiversity value to coral reefs, etc. which is not what the study shows.

7. p. 3 L. 45-46: "One or more stressors..." is vague. What kind of stressors?

8. p. 3 L. 49: "a series of consequences on multiple levels" is vague. What kind of consequences? Levels of biological organization?

9. p. 3 L. 59-60: "co-acting environmental stressors" is vague. What stressors have been suggested by the cited studies?

10. p. 4 L. 62-65: Only one publication is cited, and only in connection with one area in the Mediterranean. What are the other studies, and where else has *P. crispata* been observed in the Mediterranean?

11. p. 4 L. 63: "increase" rather than "rise"?

12. p. 4 L. 69: Are any light data available to support this statement? Light availability is not necessarily low at any given depth (in fact, co-occurrences of *P. oceanica* with *P. crispata* would suggest to me that light availability is relatively high at those locations).

13. p. 4 L. 72-74: Many studies already show algal assemblages can support high biodiversity (Miller et al. 2018 and Teagle and Smale 2018 are two recent examples, but a large number of past papers on *Macrocystis* or *Laminaria* communities would apply).

14. p. 4 L. 74-76: Confusing syntax, consider re-phrasing.

15. p. 5 L. 80, L. 85; p. 6 L. 112; p. 6 L. 126; p. 8 L. 171; p. 10 L. 220-221: Consider adding "sessile invertebrate" in these and other places in the manuscript.

16. p. 5 L. 92: "comparable to traditional biodiversity hotspots"?

17. p. 6 L. 107: The PERMANOVA gives a measure of statistical significance, but the nMDS plot by itself does not (cannot).

18. p. 6 L. 112-142: Consider revising the way the diversity analysis is presented. Some terminology here may not be clear to readers unfamiliar with Hill numbers or who are reading this section before the Methods section (e.g. "The diversity...rose and increasingly aligned in all habitats..."; the use of q and C_{max} variables; and so on).

19. p. 7 L. 149: Some more detail about what those findings were would be helpful.

20. p. 8 L. 170, p. 10 L. 213: "overlooked" rather than "overseen"

21. p. 8 L. 158-162: Perhaps "occurrence" rather than "ecological concept"? Or re-phrase this sentence? The data suggest that *P. crispata* is providing structural complexity for sessile

invertebrates that is of comparable value to seagrass beds, but I do not think it amounts to a full counterexample of the idea that “associated organisms” lose habitat with the decline of seagrass beds unless the data also include mobile invertebrates, fish, infauna, etc.

22. p. 9 L. 193-199: Confusing syntax, consider re-phrasing.

23. p. 10 L. 209: “recovering” rather than “recurring”?

24. p. 10 L. 222-224: Consider re-phrasing. I think this is an overly broad and strong statement; the authors have not yet convinced me that sessile invertebrate biodiversity is the key to restoring *P. oceanica* beds.

25. p. 11 L. 228: Can other data be provided or referenced to support the idea that the sessile invertebrate communities at these study sites do not change much annually or across seasons? Otherwise, this seems like a rather broad sampling period for direct comparisons. The biodiversity calculations depend on incidence data that might be less sensitive, but the density estimates, PERMANOVA, etc. depend on abundance data that could be quite sensitive to interannual fluctuations.

26. p. 11 L. 240: Were the 29 leaves single leaves, or sets of leaves? How many were collected at each site?

27. p. 11 L. 240 and 242: Here and elsewhere, the term used for the material other than leaves should be consistent. “Rhizomes” is used most often, but there are several places where these are referred to as “root-rhizomes” or “shoots”.

28. p. 11 L. 250: It is not clear here why the environmental parameters were being assessed.

29. p. 12 L. 264: Borg et al. 2006 is a convenient example of a paper that argues we should consider a “degraded” habitat of higher ecological value due to high biodiversity while also being clear about which facet of biodiversity is under discussion and how far they think that value goes relative to other habitats.

30. p. 14 L. 320-323: Confusing syntax, consider re-phrasing.

31. p. 15 L. 335: Citation?

32. p. 16-17, L. 374-394: I found this section to be confusingly organized. The PERMANOVA analysis (which is spelled in several different ways) seems to be introduced twice, and the rationale for using both PERMANOVA and ANOSIM is not clear. The sentence about excluding sampling location as a factor is an important point that should be elaborated upon. Analyses like PERMANOVA assume samples are independent, so ignoring the nesting of samples inside larger study sites (potential spatial autocorrelation) should be clearly justified. Modifications to Figure S4 would help (i.e. site names that match Figure S2, labels for which cluster represents which habitat).

33. p. 19 L. 442: “similar” rather than “identical”

34. p. 19 L. 426: What is a sub font?

35. p. 20 L. 450-455: This description was somewhat confusing, even with Figure S5. Does “1 cm” mean at 1 cm intervals, or 1 cm above the sediment-water interface, or something else? What adjustment was made to the *P. oceanica* setup?

36. Table 1: What does “discounting Hill numbers” mean?

37. Figure 2: I do not see brackets for unique phenotypes in *P. oceanica* compartments that are mentioned on p. 5 L. 88.

38. Table S1: Consider not including decimal places for values in this table (is 0.03 of an individual biologically meaningful here?)

39. Figure S2: I found this figure a bit confusing in combination with Figure S4 and p. 19 L. 437-438. The site names in Figure S2 and Figure S4 should fully match. If the environmental sampling location and the biodiversity sampling location both named "Punta del Morto" are separate, as the GPS coordinates suggest, consider a different name for one to distinguish them.

40. Figure S5: Were the other two clod cards in the figure not deployed for this study, or were the data simply not analyzed?

Reviewer #2 (Remarks to the Author):

Review for "Fleshy red algae mats act as reservoir for sessile invertebrate biodiversity"

General comments: This paper is on an important topic, namely, the phenomenon of macroalgal mats in seagrass meadows, which is becoming more and more frequent around the world. A lot of work has clearly gone into the study in terms of fieldwork and species identification and it has the potential for publication. However, there is a lot of improvement needed, especially in terms of framing the study in regards to existing literature, and in interpreting the results.

The introduction in particular needs a lot of improvement. There is a lot of literature, stretching back decades, on seagrass-algal mat dynamics and biodiversity of algal mats in other areas of the world, which could be discussed and cited here and would provide the background for your study as well as the hypotheses. For example:

Virnstein et al. 1985 [https://doi.org/10.1016/0304-3770\(85\)90021-X](https://doi.org/10.1016/0304-3770(85)90021-X)

Hull 1987 [https://doi.org/10.1016/0272-7714\(87\)90112-0](https://doi.org/10.1016/0272-7714(87)90112-0)

Raffaelli et al 1998 <https://www.taylorfrancis.com/chapters/edit/10.1201/b12646-13/ecological-impact-green-macroalgal-blooms-david-raffaelli-john-raven-lynda-poole>

Norkko et al. 2000 [https://doi.org/10.1016/S0022-0981\(00\)00155-6](https://doi.org/10.1016/S0022-0981(00)00155-6)

Salovius et al 2005 <https://doi.org/10.1016/j.seares.2004.05.001>

Nelson et al 2008 <https://doi.org/10.1890/07-0494.1>

Arroyo et al 2012 <https://doi.org/10.1016/j.jembe.2012.03.020>

Bohórquez et al 2013 <https://doi.org/10.1016/j.marpolbul.2013.02.002>

Arroya et al 2016 <https://doi.org/10.4324/9781315370781-6>

Coffin et al. 2017 <https://doi.org/10.7717/peerj.3080>

Barnes 2019. <https://doi.org/10.1002/aqc.2977>

(Note: this is not an exhaustive list, just the first that came to mind).

On a related note, the introduction also does not provide any motivation for the study, pose any research questions, state hypotheses or expected results. These need to be added

The conclusions of the study are also overstated, also indicating that there needs to be more framing in regards to what is already known and what this study adds to the literature.

In particular, the fact that algal mats can harbour high diversity is not really a surprise – this has been found previously. But previous studies have shown that these algal mats are not great habitat due to their transient and short-lived nature. The discussion touches upon this briefly, but if you are to propose these algal mats as reservoirs of biodiversity, there needs to be more investigation of the actual life span of these algal mats. The potential life span of this algal species is stated at a few decades, but is this true for the study area? It seems that the study deals with drifting algal mats (and I did not find this listed anywhere, so please add) in which case I have a hard time believing that they would actually provide a stable habitat for decades. In reality, they would probably be washed up during storms within a few months to a year.

In addition, the study focused mainly on the number of species. However, there is little attention on which species are found in the macroalgae vs the seagrass. Are rare species more common in the seagrass? What about endangered species? What about different taxonomic and functional

groups. If the algal mats only support common species and are missing some important taxonomic or trophic or functional groups, they are not effective reservoirs. A species list in the supplement would be helpful as well.

Based on these, toning down the conclusions is needed. Address the limitations of the study and what further information and investigations are needed to determine whether algal mats are truly effective reservoirs in this part of the world. (This applies to the discussion, abstract, and title as well).

Some more specific comments:

Introduction, general: Somewhere in the introduction or the methods, please add some general ecological information about the study species.

Results and discussion, Lines 81-84: perhaps add the geographic area where this is was studied, so that readers don't have to flip to the methods for this.

Results and discussion, Line 149: elaborate how this is line with recent findings.

Results and discussion, Lines 203-208: References needed here – there are many such references in the literature.

Methods: what the red algae sampled attached or drifting? This is likely an important distinction in terms of the longevity and stability of the habitat.

Methods: The methods are mostly very detailed and the code is available, so that is excellent! I'm not sure about the use of the word phenotypes throughout the manuscript. I understand why it was chosen, but think the word "species" or "taxa" could replace it (with a note in the methods noting that several species might be represented separately). Otherwise, the word phenotype would be more accurate to represent different types of individuals within a single species, rather than many.

Tables: Please add a species list in the supplement that shows the species found in the different habitats.

Figures: Please change the colour palette! The red and green colours used are completely indistinguishable to anybody who is colour-blind. This includes the following figures: Figure 2, Figure 3, and Figure S1

Reviewers' comments:

Reviewer #1 (Remarks to the Author):

COMMSBIO-21-2198

Fleshy red algae mats act as reservoir for sessile invertebrate biodiversity

This manuscript presents the results of a comparison of sessile invertebrate biodiversity in persistent fleshy red algae mats and *Posidonia oceanica* seagrass beds in the Mediterranean. The authors show that sessile invertebrate biodiversity is higher in the former than the latter and suggest that these algae mats may act as biodiversity reservoirs when *P. oceanica* beds are in decline or recovery. I generally agree that this type of study is important to publish, because it can provide nuance to and/or challenge paradigms that inform environmental policy decisions. However, I feel the manuscript would benefit from addressing some important points about the design, analysis, applicability, and overall message of the study.

Our response: We, the authors, highly appreciate the constructive and helpful feedback provided by the reviewer. We addressed all points raised by the reviewer. Please find detailed information further below.

General comments:

1. I think the manuscript does not emphasize enough that the study pertains to sessile invertebrate biodiversity. This is, of course, directly stated in the title, but it is glossed over in other parts of the manuscript. The result is that the manuscript both oversells the findings (by appearing to extend the findings to biodiversity too broadly) and undersells the findings (by not including much discussion of why we should care about sessile invertebrate epifauna and their effects on ecological functioning in these macroalgal and seagrass habitats, relative to other groups that have received more attention). I think adjustments are advisable throughout the manuscript – some specific suggestions are below – but are especially important in the abstract.

Our response: The manuscript has been revised accordingly and 'invertebrate diversity' has been changed to 'sessile invertebrate diversity' where necessary. Overall, we have toned down the results in the revised manuscript by clearly stating that the results of the present study apply to sessile invertebrate biodiversity, and that it is known that algal habitats may host high associated biodiversity. We emphasised findings of the present study by addressing the role of *P. crispata* as an ecosystem engineer, and how this could favour the association with certain sessile invertebrates. By clarifying the conducted biodiversity analysis we highlight that our biodiversity analysis particularly considers the occurrence of highly frequent, frequent and rare species (via the concept of Hill numbers). The results sections have been edited to include further details on the occurrence of rare species associated with *P. crispata* vs. *P. oceanica*. The abstract has been edited accordingly. We refer to the point-to-point response below for further detailed information.

2. It would be helpful for the introduction to justify why sessile invertebrate epifauna were specifically targeted for this study. p. 4 L. 75 suggests that *P. crispata* is a general diversity hotspot, but the cited study is specific to serpulid polychaetes. Seagrass beds are known for hosting diverse assemblages of many functional groups: not only sessile invertebrate epifauna, but also epiphytes, mobile invertebrate epifauna, fish, meioinfauna, and macroinfauna. Among these groups, why were sessile invertebrate epifauna chosen for special attention here? Was sampling for mobile invertebrate epifauna considered? Mobile invertebrate epifauna have been included in numerous past comparisons of seagrasses and macroalgae, including the Thomsen et al. 2012 study cited in this manuscript to justify potentially considering *P. crispata* a reservoir.

Our response: The introduction has been clarified accordingly. We agree to the reviewer that mobile invertebrates provide an essential part of the invertebrate biodiversity, as previously shown in numerous studies. In the present study, we focused on the sessile benthic invertebrates, because a) the pilot studies by Bonifazi et al. (2017) and Roszbach et al. (2021) indicated high related diversity, b) the sampling procedure could only ensure complete retrieval of sessile, but not mobile invertebrates, c) this group also reflects the stability and longevity of the red algae mats as habitat. We have now added this information in the revised manuscript.

Edit in the manuscript:

I. 83-94: “But, it remains largely unknown whether the mat-forming red alga *P. crispera* that is likely increasing in abundance and potentially replaces classical high biodiversity habitats such as seagrass meadows and rocky substrates harbours high associated sessile biodiversity. Thus, we here aimed to determine the role *P. crispera* as habitat for sessile invertebrate biodiversity. This relates to recent pilot studies that have identified non-colonial²² and sessile polychaetes²⁹ to be associated with *P. crispera* mats. The present study, hence, aims to answer the following research questions: i) to what extend can *P. crispera* mats function as habitat for sessile invertebrates, and ii) how does this biodiversity compare to neighbouring *Posidonia oceanica* seagrass meadows? Based on aforementioned indications^{22,29}, we hypothesise that *P. crispera* harbours high associated sessile invertebrate diversity. Furthermore, we expect that this biodiversity potentially differs from that of *P. oceanica* due to different habitat characteristics.”

3. The authors' argument that red algae mats fulfill ecosystem engineering functions could be strengthened by providing further justification for the methods, and by putting the results more fully in context of the other ecosystem engineers to which the algae are being compared (here, seagrasses, specifically *P. oceanica*) and the organisms benefitting from their modification of the environment (here, sessile invertebrates). First, is there additional support for the idea that 5 days of environmental data from a single site are representative of most or all of the other sites examined? Second, can more detail about how similarity between habitats was assessed be provided? Dissolved oxygen and pH, for example, may differ between seagrass beds and macroalgae mats, but the difference is often in degree of fluctuation rather than level per se (e.g. Viaroli et al. 2008), and this may or may not be obvious in simple plots of the data. At the very least, perhaps the cited findings in Schmidt et al. 2021 could be elaborated upon. Third, seagrasses affect their environment in ways other than the parameters listed, such as stabilizing sediment and profoundly influencing geochemical fluxes in the water column and sediment. Not mentioning any of these makes sense only if the focus is clearly on sessile invertebrate epifauna throughout the manuscript.

Our response: We appreciate the points raised by the reviewer. Concerning the three mentioned points:

- 1) Each site was assessed for a total of 10 to 18 days. The method section has been edited to state this more clearly.
- 2) Mentioned studies and respective findings have been included and discussed in the present manuscript.
- 3) We agree that sediment trapping and resulting influences on biogeochemical fluxes are worth mentioning in the context of the present manuscript. A discussion about this topic has been added to the manuscript.

Edits in the manuscript:

I. 556-557: “Manta loggers were deployed multiple times (13x in *P. crispa* mats, 7x in *P. oceanica* meadows, 6x in reference habitat) for 2 to 3 days.”

I. 565-567: “All setups were assembled prior to deployment and were positioned multiple times (9x in *P. crispa* mats, 4x in *P. oceanica* meadows, 2x in reference habitat) for 6 to 7 days in the respective habitat.”

I. 573-578: “Similar setups were prepared for light intensity and temperature assessments. Instead of GCC, Onset HOBO Pendant Data Loggers (part #UA-002-64) were placed multiple (10x in *P. crispa* mats, 6x in *P. oceanica* meadows, 4x in reference habitat) times accordingly recording at 15 sec intervals for 5 days in a row. A total of 10 to 18 days and respective data points covering each 15 seconds (in case of Onset HOBO Pendant Data Loggers) or each minute (in case of Eureka Manta Loggers) of a daily cycle were collected.”

I. 202-208: “Daily oxygen concentration fluctuations of *P. crispa* (7.73 – 8.14 mg l⁻¹) were similar to those of *P. oceanica* (7.59 – 8.04 mg l⁻¹), with the daily mean of oxygen concentrations being slightly higher in *P. crispa* (7.99 mg l⁻¹) compared to those of *P. oceanica* (7.75 mg l⁻¹). This contrasts previous findings stating that shallow, macroalgae-covered environments undergo wider oxygen concentration fluctuations compared to seagrass meadows^{47,48}. Our findings indicate that this is not necessarily the case in deeper environments (Supplementary Fig. S1).“

I. 217-224: “The extent to which further functions such as sediment trapping, related to reduced water movements induced by *P. oceanica* meadows⁵⁶⁻⁵⁸, apply to *P. crispa* mats need to be determined. The complex morphology of *P. crispa* mats, reflected by the surface area enlargement (see next paragraph) and a high surface area (provided by the complexity of the thalli) relative to a small volume (mats of several cm thickness), suggest that sediment and particulate matter are likely trapped. Trapped sediment and particulate matter, hence, could provide a) a heterogeneous habitat for infaunal species⁵⁹ and, b) (in-)organic matter for tube-building species such as sessile polychaetes⁵⁹.“

4. More context for the authors’ suggestion that *P. crispa* mats may act as reservoirs of sessile invertebrate biodiversity would be helpful. Are there groups of sessile invertebrates in *P. crispa* mats that the authors feel would particularly assist in re-establishing *P. oceanica* beds or their functions (e.g. filter feeders that help to maintain water clarity)? A discussion earlier in the results of which sessile invertebrate phenotypes or taxa are responsible for distinguishing the three habitats examined would also be useful – I think the argument that *P. crispa* mats can act as reservoirs for *P. oceanica* because their community taxa overlap so much is blunted in the text and Figure 2 by the very clear distinction in community compositions from the nMDS analysis.

Our response: We would like to emphasise that it was our postulation that *P. crispa* could function as a temporary reservoir for sessile invertebrates. Nevertheless, we agree with the reviewer that even though both *P. crispa* and *P. oceanica* harbour a range of unique phenotypes, there is a considerable overlap of shared phenotypes as shown in the Venn diagram (Fig. 2A). It remains speculative to what extent the initial sessile invertebrate community composition of *P. oceanica* may be re-colonised by communities in *P. crispa* mats. The mentioned distinction in the community composition was shown in the nMDS analysis and plot (Fig. 2B). This point has been added to the discussion.

Edits in the manuscript:

I. 297-308: “In the Mediterranean, we postulate that *P. crispata* can support *P. oceanica* meadows (and other habitats, see Table 1) by maintaining their sessile invertebrate biodiversity^{62,72}, particularly due to the considerable overlap of shared phenotypes, i.e., sessile invertebrates that occur in both *P. crispata* and *P. oceanica* (Fig. 2A), even though both habitats harbour a range of unique phenotypes. It nevertheless remains to be determined to what extent the community composition in re-colonised *P. oceanica* meadows differs from their initial sessile invertebrate community composition, considering the clear distinction of associated sessile invertebrate communities in *P. crispata* mats and *P. oceanica* meadows (Fig. 2B). Since marine species and their habitats are capable of recovery once threats are reduced or removed^{19,69,71}, our findings suggest that *P. crispata* mats and their associated sessile invertebrate communities potentially aid reviving classical marine (sessile invertebrate) biodiversity hotspots not only in the Mediterranean.”

5. I would urge the authors to carefully consider the clarity and flow of their writing during the revision process, especially in the Methods section (and in consideration of the fact that the Results and Discussion section will be read before the Methods section).

Our response: Clarity and flow of the manuscript have been checked carefully and improved by all co-authors. We refer to the specific comments by the reviewer and reviewer #2 for further information.

Specific comments:

6. p. 2 L. 29-32: I recommend including the words sessile and/or sessile invertebrate somewhere in these concluding sentences (e.g. “alternative habitats and sessile invertebrate biodiversity reservoirs...”). If the reader misses the “sessile” on line 28, the last two sentences of the abstract imply that these red algae mats are habitats of equivalent general biodiversity value to coral reefs, etc. which is not what the study shows.

Our response: Has been edited according to the reviewers’ suggestion.

Edits in the manuscript:

I. 32-34: “Our findings suggest that fleshy red algae mats act as alternative habitats and temporary sessile invertebrate biodiversity reservoirs in times of environmental change.”

7. p. 3 L. 45-46: “One or more stressors...” is vague. What kind of stressors?

Our response: Further information has been added.

Edits in the manuscript:

I. 47-49: “One or more stressors, e.g., ocean warming⁶ and acidification¹² or nutrient pollution⁶, can alter the community dynamics, shifting the system to alternative states dominated by more tolerant species^{6,12}.”

8. p. 3 L. 49: “a series of consequences on multiple levels” is vague. What kind of consequences? Levels of biological organization?

Our response: In this context, a broad range of consequences was meant. Commonly, these series of consequences entail a loss of structural and spatial complexity, and hence, biodiversity. Additional information has been added to the manuscript.

Edits in the manuscript:

I. 52-54: “Such shifts naturally entail a series of consequences on multiple levels, such as loss of ecosystem services and functioning^{11,14}, and are usually associated with a loss of structural/spatial complexity, and consequently, biodiversity^{3,6,15,16}.”

9. p. 3 L. 59-60: “co-acting environmental stressors” is vague. What stressors have been suggested by the cited studies?

Our response: Cited studies refer to habitat loss and degradation, pollution, eutrophication or ocean warming. This information has been added to the manuscript.

Edits in the manuscript:

I. 62-66: “Likewise, co-occurring environmental stressors such as habitat loss and degradation¹⁸, pollution^{5,18}, eutrophication^{5,18} or ocean warming¹⁸, promoted the formation of persistent²¹, turf- and mat-forming algal assemblages of the species *Phyllophora crispa* (formerly *P. nervosa*²²) that have been observed across the Mediterranean^{23,24}, the Black Sea^{25,26} and the Atlantic^{24,27}.”

10. p. 4 L. 62-65: Only one publication is cited, and only in connection with one area in the Mediterranean. What are the other studies, and where else has *P. crispa* been observed in the Mediterranean?

Our response: To avoid misunderstandings, further studies have been added to the sentence.

Edits in the manuscript:

I. 66-70: “A growing number of publications addressing *P. crispa* suggests an increase of these algae in the Mediterranean^{22,23,28,29}: *P. crispa* has been observed along the coast of Sardinia, Italy²³ and lately in the Tyrrhenian Sea, Italy, for the first time²², where it forms dense mats of up to 15 cm thickness (Fig. 1A).”

11. p. 4 L. 63: “increase” rather than “rise”?

Our response: Has been edited accordingly.

Edits in the manuscript:

I. 66-68: “A growing number of publications addressing *P. crispa* suggests an increase of these algae in the Mediterranean^{22,23,28,29}.”

12. p. 4 L. 69: Are any light data available to support this statement? Light availability is not necessarily low at any given depth (in fact, co-occurrences of *P. oceanica* with *P. crispa* would suggest to me that light availability is relatively high at those locations).

Our response: We agree with the point raised by the reviewer. We have, thus, edited the sentence to avoid misunderstandings. Further, related information about the light availability in *P. crispa* and *P. oceanica* quantified for the present study has been added to the results and discussion section, and can be found in Supplementary Fig. S1.

Edits in the manuscript:

I. 73-74: “*P. crispa* is sciaphilic^{22,26}, i.e., adapted to low-light conditions, and reaches large accumulations in water depths between 10 and 55 m^{25,26}.”

I. 196-202: “Our results show that *P. crispata* shapes key environmental parameters similarly to neighbouring *P. oceanica* seagrass meadows (Supplementary Fig. S1), as particularly water movement and light intensity within the red algae mats as well as in the seagrass meadows are reduced compared to neighbouring bare substrate (Supplementary Fig. S1B, F). This confirms findings of a parallel study that has identified *P. crispata* as an ecosystem engineer modifying its environment²⁸.”

13. p. 4 L. 72-74: Many studies already show algal assemblages can support high biodiversity (Miller et al. 2018 and Teagle and Smale 2018 are two recent examples, but a large number of past papers on *Macrocystis* or *Laminaria* communities would apply).

Our response: We agree to the point raised by the reviewer, and added the suggestions to the manuscript.

Edits in the manuscript:

I. 77-86: “Some previous studies identified a high biodiversity associated with drifting algae in a lagoon off the west coast of the United States³⁰ and in the Baltic Sea³¹⁻³³. Calcifying green algae communities in coral reefs of the Great Barrier Reef, Australia³⁴, and green algae occurring in blooms in the United States³⁵, Canada³⁶ and South Africa³⁷, as well as at further locations of the Atlantic^{38,39}, furthermore, indicate that algal assemblages can support high biodiversity. Additionally, kelp-forming brown algae in the United States⁴⁰ and United Kingdom⁴¹ host a vast array of associated organisms. But, it remains largely unknown whether the mat-forming red alga *P. crispata* that is likely increasing in abundance and potentially replaces classical high biodiversity habitats such as seagrass meadows and rocky substrates harbours high associated sessile biodiversity.”

14. p. 4 L. 74-76: Confusing syntax, consider re-phrasing.

Our response: Has been rephrased.

Edits in the manuscript:

I. 86-94: “Thus, we here aimed to determine the role *P. crispata* as habitat for sessile invertebrate biodiversity. This relates to recent pilot studies that have identified non-colonial²² and sessile polychaetes²⁹ to be associated with *P. crispata* mats. The present study, hence, aims to answer the following research questions: i) to what extent can *P. crispata* mats function as habitat for sessile invertebrates, and ii) how does this biodiversity compare to neighbouring *Posidonia oceanica* seagrass meadows? Based on aforementioned indications^{22,29}, we hypothesise that *P. crispata* harbours high associated sessile invertebrate diversity. Furthermore, we expect that this biodiversity potentially differs from that of *P. oceanica* due to different habitat characteristics.”

15. p. 5 L. 80, L. 85; p. 6 L. 112; p. 6 L. 126; p. 8 L. 171; p. 10 L. 220-221: Consider adding “sessile invertebrate” in these and other places in the manuscript.

Our response: Has been added accordingly to the mentioned lines and in further parts of the manuscript.

16. p. 5 L. 92: “comparable to traditional biodiversity hotspots”?

Our response: Has been edited accordingly.

Edits in the manuscript:

I. 110-112: “Calculation of classical diversity indices further endorsed *P. crispata* as a hotspot of sessile invertebrate diversity comparable to traditional biodiversity hotspots

such as coral reefs or Mediterranean coralligenous reefs (Table 1).”

17. p. 6 L. 107: The PERMANOVA gives a measure of statistical significance, but the nMDS plot by itself does not (cannot).

Our response: Has been clarified.

Edits in the manuscript:

I. 126-131: “The nMDS plot and appendant statistical analysis revealed that sessile invertebrate communities significantly varied among habitats (PERMANOVA with all $p < 0.001$), regardless of numbers of phenotypes and individuals of the investigated habitats (ANOSIM: *P. crispa* – *P. oceanica* leaves: $R = 1$; *P. crispa* – *P. oceanica* rhizomes: $R = 1$; *P. oceanica* leaves – rhizomes: $R = 0.98$; incidence data and Bray Curtis similarities).”

18. p. 6 L. 112-142: Consider revising the way the diversity analysis is presented. Some terminology here may not be clear to readers unfamiliar with Hill numbers or who are reading this section before the Methods section (e.g. “The diversity...rose and increasingly aligned in all habitats...”; the use of q and C_{max} variables; and so on).

Our response: We acknowledge the point raised by the reviewer. We have revised the respective section thoroughly and introduced key terminology in the main text of the manuscript.

Edits in the manuscript:

I. 134-191: “This was done to account for differences in sampling efforts, i.e., number of samples collected per habitat. Hill numbers represent the effective number of equally abundant species ${}^qD^{43,44}$, where q denotes the diversity order of a Hill number. The parameter q determines the sensitivity to species’ frequencies and Hill numbers based on increasing values of q to emphasize on more frequently occurring species. In our analysis, qD of orders $q = 0$, $q = 1$, and $q = 2$ were calculated, representing phenotype richness (i.e., phenotypes quantified equally disregarding frequency, 0D), Shannon diversity (i.e., effective number of frequent phenotypes, 1D) and Simpson diversity (i.e., effective number of highly frequent phenotypes, 2D), respectively ⁴⁵ (see Methods for further details).

The estimated sample completeness (i.e., diversity detected) profiles implied that there was undetected diversity within the habitats (Fig. 3A). Sample completeness profiles revealed that between 73.0 % (*P. oceanica* holobiont) and 85.7 % (*P. crispa*) of phenotype richness ($q = 0$) was detected with no significant differences among (sub-) habitats (i.e., respective 95 % confidence intervals overlapped). The diversity detected in the (sub-) habitats rose with order q (i.e., diversity detected increased for more frequently occurring species) and increasingly aligned in all habitats for Shannon ($q = 1$) and Simpson ($q = 2$) diversity, with the majority of frequent and highly frequent phenotypes being detected (Fig. 3A, Supplementary Table S3). To test if we could reliably estimate diversity based on our data, sample-size-based rarefaction and extrapolation curves were computed to check for asymptoted values of q . An estimation of true Simpson diversity based on our data for all (sub-) habitats was indeed reliable (i.e., size-based rarefaction and extrapolation curves asymptoted for $q = 2$; Fig. 3B). Hence, we could confirm that *P. crispa* harbours assemblages with significantly higher Simpson diversity (~132; Fig 3C, Supplementary Table S3; no overlap of 95% confidence intervals ⁴⁶) compared to all other (sub-) habitats, emphasising its role as a biodiversity hotspot for sessile invertebrates. For phenotype richness and Shannon diversity, only conservative minimum estimates could be

obtained, as size-based rarefaction and extrapolation curves did not asymptote for $q = 0,1$ (Fig. 3B). In this case, statistically reliable comparison between habitats' phenotype richness and Shannon diversity may only be performed based on standardised data. For this purpose, we compared diversities based on standardised data at a sample coverage level of $C_{\max} = 96.9\%$ (Fig. 3D, Supplementary Table S3). C_{\max} is the lowest sample completeness at $q = 1$ of any (sub-) habitat when samples are extrapolated to double the respective number of samples per (sub-) habitat. Consequently, we showed that *P. crispa* exhibited significantly (i.e., no overlap of respective 95 % confidence intervals⁴⁶; Fig. 3D, Supplementary Table S3) higher phenotype richness compared to neighbouring *P. oceanica*: phenotype richness of *P. crispa* mats (~234 phenotypes) exceeded those of *P. oceanica* rhizomes (~142) and leaves (~102) at a fixed sample coverage of $C_{\max} = 96.9\%$ (Fig. 3D, Supplementary Table S3⁴⁶), whereas the difference compared to *P. oceanica* holobiont (~207) was marginal. For Shannon diversity, *P. crispa* showed a significantly higher index value (~159) compared to the *P. oceanica* holobiont (~111), leaves (~64) and rhizomes (~84; see Fig 3D, Supplementary Table S3⁴⁶). Phenotype evenness (i.e., Pielou's J at C_{\max} ; an evenness measure based on phenotype occurrences) was relatively high for all habitats, being lowest for the *P. oceanica* holobiont (0.88) and highest for *P. crispa* (0.93; Supplementary Table S3). Furthermore, *P. crispa* harboured the most evenly diverse biotic communities among all (sub-) habitats for all orders of $q > 0$ at C_{\max} (i.e., for orders of increasing sensitivity to phenotype frequencies; Fig. 3E; Supplementary Table S3). The difference in estimated phenotype Simpson diversity between *P. crispa* and the *P. oceanica* holobiont at C_{\max} is larger than the difference in phenotype richness (i.e., ~50 and ~27, respectively; Supplementary Table S3). Comparing the empirical richness values (i.e., values for $q = 0$) with the values estimated asymptotically and non-asymptotically (the latter standardised for C_{\max}), it is shown that the number of undetected phenotypes is larger for (sub-) habitats of *P. oceanica* (holobiont and rhizomes) than for *P. crispa* (Supplementary Table S3). These findings indicate that the higher overall diversity in *P. crispa* may be driven by the higher abundance of frequently occurring phenotypes rather than rare phenotypes, while *P. oceanica* seems to harbour a higher number of undetected rare phenotypes. Conclusively, our data allowed us to identify *P. crispa* as a valuable habitat harbouring more even and diverse sessile invertebrate communities that are shaped by the higher abundance of frequently occurring phenotypes compared to neighbouring *P. oceanica* meadows, which in return harbours more undetected rare phenotypes.”

19. p. 7 L. 149: Some more detail about what those findings were would be helpful.

Our response: Further information has been added to the manuscript. Now, detailed information about the environmental parameters and the link between those and assessed sessile biodiversity is discussed.

Edits in the manuscript:

I. 196-217: “Our results show that *P. crispa* shapes key environmental parameters similarly to neighbouring *P. oceanica* seagrass meadows (Supplementary Fig. S1), as particularly water movement and light intensity within the red algae mats as well as in the seagrass meadows are reduced compared to neighbouring bare substrate (Supplementary Fig. S1B, F). This confirms findings of a parallel study that has identified *P. crispa* as an ecosystem engineer modifying its environment²⁸. This function also applies to further environmental parameters: Daily oxygen concentration fluctuations of *P. crispa* (7.73 – 8.14 mg l⁻¹) were similar to those of *P. oceanica* (7.59

– 8.04 mg l⁻¹), with the daily mean of oxygen concentrations being slightly higher in *P. crispera* (7.99 mg l⁻¹) compared to those of *P. oceanica* (7.75 mg l⁻¹). This contrasts previous findings stating that shallow, macroalgae-covered environments undergo wider oxygen concentration fluctuations compared to seagrass meadows^{47,48}. Our findings indicate that this is not necessarily the case in deeper environments (Supplementary Fig. S1). Additionally, pH within *P. crispera* was lower (8.44) compared to *P. oceanica* meadows (8.64). Shaped environmental parameters, particularly reduced water movement and potentially varying oxygen availability, could benefit the settlement of specific bryozoan^{49–53}, bivalve⁵⁴ and polychaete⁵³ species. This may explain findings of the present study, in which we identified more phenotypes and/or individuals associated with *P. crispera* compared to *P. oceanica* of bryozoans (76 vs. 78 phenotypes, 44222 vs. 7655 indiv. per m² habitat), molluscs (bivalves; 4 vs. 4 phenotypes, 112 vs. 38 indiv. per m² habitat) and polychaetes (23 vs. 13 phenotypes, 5950 vs. 3734)⁵⁵ associated with *P. crispera* compared to *P. oceanica* (see Supplementary Tab. S1).“

20. p. 8 L. 170, p. 10 L. 213: “overlooked” rather than “overseen”

Our response: Has been edited accordingly.

Edits in the manuscript:

I. 242-246: “Taken together, the results of the comparative biodiversity indices (Table 1), biodiversity analysis (Fig. 2, 3), and environmental parameters shaped by *P. crispera* (Supplementary Fig. S1) along with the considerable surface area enlargement (Supplementary Table S2) and its wide distribution^{22,24,26,27} show that persistent red algae mats could function as overlooked ecosystem engineers that harbour high sessile invertebrate biodiversity.”

21. p. 8 L. 158-162: Perhaps “occurrence” rather than “ecological concept”? Or re-phrase this sentence? The data suggest that *P. crispera* is providing structural complexity for sessile invertebrates that is of comparable value to seagrass beds, but I do not think it amounts to a full counterexample of the idea that “associated organisms” lose habitat with the decline of seagrass beds unless the data also include mobile invertebrates, fish, infauna, etc.

Our response: We agree with the point raised by the reviewer and followed the suggestion. We furthermore added another reference in this context.

Edits in the manuscript:

I. 233-236: “Growth form, enlargement factor and persistence²⁵ of *P. crispera* contradict the common occurrence that structural complexity is drastically reduced when spatially complex and long-living habitats, such as seagrass meadows, decline^{2,6}.”

22. p. 9 L. 193-199: Confusing syntax, consider re-phrasing.

Our response: Has been rephrased.

Edits in the manuscript:

I. 267-273: “In sharp contrast to *P. oceanica* meadows, *P. crispera* mats, particularly those with an unattached form growing on sediments²⁶, are more fragile and can be dislodged and translocated by waves²⁵. Presumably, the translocation of *P. crispera* mats has consequences for associated biodiversity: i) translocated *P. crispera*²⁵, which can colonise and spread vegetative, still provides habitat for associated sessile invertebrates; or ii) *P. crispera* mats are severely damaged, losing their function as

ecosystem engineers, and, hence, biodiversity hotspots.”

23. p. 10 L. 209: “recovering” rather than “recurring”?

Our response: Has been edited accordingly.

Edits in the manuscript:

I. 287-288: “We postulate that sessile invertebrates can re-colonise recovering *P. oceanica* meadows, in case appropriate conservation measures are successfully implemented^{17,68}.”

24. p. 10 L. 222-224: Consider re-phrasing. I think this is an overly broad and strong statement; the authors have not yet convinced me that sessile invertebrate biodiversity is the key to restoring *P. oceanica* beds.

Our response: Has been toned down and rephrased.

Edits in the manuscript:

I. 297-308: “In the Mediterranean, we postulate that *P. crispa* can support *P. oceanica* meadows (and other habitats, see Table 1) by maintaining their sessile invertebrate biodiversity^{62,72}, particularly due to the considerable overlap of shared phenotypes, i.e., sessile invertebrates that occur in both *P. crispa* and *P. oceanica* (Fig. 2A), even though both habitats harbour a range of unique phenotypes. It nevertheless remains to be determined to what extent the community composition in re-colonised *P. oceanica* meadows differs from their initial sessile invertebrate community composition, considering the clear distinction of associated sessile invertebrate communities in *P. crispa* mats and *P. oceanica* meadows (Fig. 2B). Since marine species and their habitats are capable of recovery once threats are reduced or removed^{19,69,71}, our findings suggest that *P. crispa* mats and their associated sessile invertebrate communities potentially aid reviving classical marine (sessile invertebrate) biodiversity hotspots not only in the Mediterranean.”

25. p. 11 L. 228: Can other data be provided or referenced to support the idea that the sessile invertebrate communities at these study sites do not change much annually or across seasons? Otherwise, this seems like a rather broad sampling period for direct comparisons. The biodiversity calculations depend on incidence data that might be less sensitive, but the density estimates, PERMANOVA, etc. depend on abundance data that could be quite sensitive to interannual fluctuations.

Our response: The data used for the present manuscript was generated between May and July 2019. This information has been included in the revised manuscript.

Edits in the manuscript:

I. 312-314: “All data were generated between May and July 2019 along the north-eastern and north-western coasts of Giglio Island, within the Tuscan Archipelago National Park, Tyrrhenian Sea, Italy (Supplementary Fig. S2), by SCUBA diving.”

26. p. 11 L. 240: Were the 29 leaves single leaves, or sets of leaves? How many were collected at each site?

Our response: A total of 29 single leaves were sampled. Information about the sampled number of specimens for *P. crispa*, *P. oceanica* rhizomes and leaves at each location was added to the method section and the supplementary (see Supplementary Fig. S2).

Edits in the manuscript:

I. 346-352: “A total of 20 *P. oceanica* rhizome specimens and 29 single leaves were collected from the four sampling sites to minimise the impact on threatened *P. oceanica* meadows. The number of sampled specimens at the respective sampling location were 4x “Corvo”, 4x “Fenaio”, 4x “Punta del Morto”, and 4x “Secca 2” for *P. crispata*; 10x “3 Fratelli”, 5x “Fenaio”, 5x “Cala Calbugina”, and 9x “Secca 2” for *P. oceanica* leaves; 10x “3 Fratelli” and 10x “Secca 2” for *P. oceanica* rhizomes (see Supplementary Fig. S2).“

27. p. 11 L. 240 and 242: Here and elsewhere, the term used for the material other than leaves should be consistent. “Rhizomes” is used most often, but there are several places where these are referred to as “root-rhizomes” or “shoots”.

Our response: We have clarified the term “rhizomes” and used it consistently henceforth. Has been edited accordingly.

Edit in the manuscript:

I. 343-345: “*P. oceanica* root-rhizomes were cut including the sheaths, both vertical and horizontal rhizome as well as the upper layers of the roots (Supplementary Fig. S3, hereafter referred to as “*P. oceanica* rhizome”).“

28. p. 11 L. 250: It is not clear here why the environmental parameters were being assessed.

Our response: We agree to the reviewer that the information at this point of the manuscript is redundant. Has been edited accordingly.

29. p. 12 L. 264: Borg et al. 2006 is a convenient example of a paper that argues we should consider a “degraded” habitat of higher ecological value due to high biodiversity while also being clear about which facet of biodiversity is under discussion and how far they think that value goes relative to other habitats.

Our response: We agree to the point raised by the reviewer. As the chosen reference does not completely support the given statement, we referred to the original references included in Borg et al. (2006).

30. p. 14 L. 320-323: Confusing syntax, consider re-phrasing.

Our response: Has been rephrased.

Edits in the manuscript:

I. 426-428: “We performed the aforementioned workflow to nonetheless achieve meaningful comparisons of investigated biotic communities according to the following steps (see Chao et al. (2020)⁴⁵ and Supplementary Table S3):”

31. p. 15 L. 335: Citation?

Our response: Citation for the Hill number diversity estimation procedure [I]-IV] is given in line 428 (Chao et al. 2020). For the particular case mentioned here, Chao et al. (2020) in turn refer to Colwell et al. (2012) in their paper. We now have included the latter as reference for this particular statement in our manuscript.

Edits in the manuscript:

I. 438-440: “Extrapolation was performed to double the actual number of samples per habitat, as further extrapolation is unreliable in the case of phenotype richness⁸⁴.“

32. p. 16-17, L. 374-394: I found this section to be confusingly organized. The PERMANOVA analysis (which is spelled in several different ways) seems to be introduced twice, and the rationale for using both PERMANOVA and ANOSIM is not clear. The sentence about excluding sampling location as a factor is an important point that should be elaborated upon. Analyses like PERMANOVA assume samples are independent, so ignoring the nesting of samples inside larger study sites (potential spatial autocorrelation) should be clearly justified. Modifications to Figure S4 would help (i.e. site names that match Figure S2, labels for which cluster represents which habitat).

Our response: The introduction and spelling of PERMANOVA has been corrected. Figure S4 has been modified according to the reviewers' suggestions and now includes matching site names and labels for the clusters that give information about the respective habitat.

Edits in the manuscript:

I. 479-499: “Non-parametric permutational multivariate analysis of variance (PERMANOVA⁹¹; based on species abundance data using Primer-E v6⁹² with the PERMONVA+ extension)⁹³ was used to check for significant differences in the sessile invertebrate community composition among (sub-) habitats. For this, raw count data (related to habitat m²) were square-root transformed to generate Bray-Curtis similarity matrices for PERMANOVAs with habitats as a factor. Pair-wise PERMOVA tests were then performed with unrestricted permutation of raw data (999 permutations), Type III (partial) sum of squares and Monte Carlo tests. In case pair-wise comparisons exhibited significant differences, we checked if these differences may partially or fully be driven by heterogeneity of multivariate dispersion. In addition, differences in the sessile invertebrate community composition were visualised applying non-metric multidimensional scaling (nMDS) based on Bray-Curtis similarities. To exclude the parameter ‘sampling location’ as a major driver shaping biodiversity patterns, a nMDS plot based on Bray-Curtis similarities was performed (see Supplementary Fig. S4). An analysis of similarities (ANOSIM) and PERMANOVA were performed based on the similarity calculations and on Bray Curtis similarities (incidence data), in order to test for differences between (sub-) habitats. The ANOSIM global R was calculated as the difference of between-tested-groups and within-tested-groups mean rank similarities, and thus it displays the degree of separation between them. Complete separation is indicated by R = 1, whereas R = 0 suggests no separation⁹². Finally, an area-proportional Venn diagram was constructed to describe the shared and unique phenotypes among (sub-)habitats, i.e., *P. crispa* mats, *P. oceanica* leaves and *P. oceanica* rhizomes.”

Supplementary Fig. S4:

33. p. 19 L. 442: “similar” rather than “identical”

Our response: Has been edited accordingly.

Edits in the manuscript:

I. 550-551: “Thus, all habitats likely experienced similar environmental conditions allowing a direct comparison of environmental parameters between the habitats.”

34. p. 19 L. 426: What is a sub font?

Our response: We intended to say “underground” or “sample surface”. Has been edited in the manuscript.

Edits in the manuscript:

I. 532-534: “The chains served to trace the actual dimensions (diagonals and edges) of the underlying substrate enclosed by the projection of the frame’s planar dimensions onto the sample surface.”

35. p. 20 L. 450-455: This description was somewhat confusing, even with Figure S5. Does “1 cm” mean at 1 cm intervals, or 1 cm above the sediment-water interface, or something else? What adjustment was made to the *P. oceanica* setup?

Our response: We agree with the reviewer that the description is confusing. Clod cards were deployed 1 cm above the seafloor within the red algae mats and 20 cm above the mats to address water movement within and outside the mats. Accordingly, clod cards were deployed at 20 cm above the seafloor within the seagrass meadows as well as 20 cm above the seagrass meadow. The section has been edited and rephrased in the manuscript.

Edits in the manuscript:

I. 560-565: “GCC were placed at 1 cm above the seafloor (i.e., within the mat) and 20 cm above the mat height attached to a metal stick with a 90° offset (Supplementary Fig. S5) to address water movement within and above *P. crispata* mats. An identical GCC setup was used on hard bottom substrates as a reference. Due to differences in height, the setup was adjusted for *P. oceanica* meadows, i.e., GCCs were placed at 20

cm above the seafloor within the meadows as well as 20 cm above the meadows.”

36. Table 1: What does “discounting Hill numbers” mean?

Our response: We acknowledge the point raised by the reviewer that this information is misleading. Hence, we edited this part as follows:

I. 900-901: “Note: indices and evenness presented here were calculated based on classical formulas and not based on Hill number calculations to enable comparison with literature data (see Methods).”

37. Figure 2: I do not see brackets for unique phenotypes in *P. oceanica* compartments that are mentioned on p. 5 L. 88.

Our response: Figure has been edited accordingly and now includes information about the unique phenotypes in *P. oceanica* leaves and rhizomes. Note: the colour palette has changed here and throughout the manuscript.

Edits in the manuscript:

38. Table S1: Consider not including decimal places for values in this table (is 0.03 of an individual biologically meaningful here?)

Our response: We acknowledge the point raised by the reviewer. We considered to not include decimal places in this table and throughout the manuscript. However, the values included in table S1 and throughout the manuscript are calculated values, based on rarefaction/extrapolation and related to seafloor or habitat surface area. Hence, despite potentially not being biologically meaningful, we kept decimal spaces to be as precise as possible in table S1, but edited throughout the manuscript to improve readability.

39. Figure S2: I found this figure a bit confusing in combination with Figure S4 and p. 19 L. 437-438. The site names in Figure S2 and Figure S4 should fully match. If the environmental sampling location and the biodiversity sampling location both named “Punta del Morto” are separate, as the GPS coordinates suggest, consider a different name for one to distinguish them.

Our response: We agree to the points raised by the reviewer. Site names in Figures S2 and S4 now match. Furthermore, the sampling location for the environmental data was close to the dive site ‘Punta del Morto’ as stated in the method section. We, hence, refrained to edit the Figure as its solely descriptive for the biodiversity assessment and refer to the GPS coordinates given in the method section.

40. Figure S5: Were the other two clod cards in the figure not deployed for this study, or were the data simply not analyzed?

Our response: All four clod cards were deployed; however, data of the respective clod cards was not considered for this study but was included in Schmidt et al. (2021, *Frontiers in Marine Science*). Information has been added to the caption.

Edits in the manuscript:

Figure S 5: “Schematic figure of utilised gypsum clod card setup; data acquired for the present study based on indicated gypsum clod cards. For further information on water movement based on this setup, we refer to Schmidt et al. (2021)¹⁷.”

Reviewer #2 (Remarks to the Author):

Review for “Fleshy red algae mats act as reservoir for sessile invertebrate biodiversity”

General comments: This paper is on an important topic, namely, the phenomenon of macroalgal mats in seagrass meadows, which is becoming more and more frequent around the world. A lot of work has clearly gone into the study in terms of fieldwork and species identification and it has the potential for publication. However, there is a lot of improvement needed, especially in terms of framing the study in regards to existing literature, and in interpreting the results.

Our response: We are very grateful for the constructive and encouraging feedback you provided. We addressed all points raised, please see below for more detailed information.

The introduction in particular needs a lot of improvement. There is a lot of literature, stretching back decades, on seagrass-algal mat dynamics and biodiversity of algal mats in other areas of the world, which could be discussed and cited here and would provide the background for your study as well as the hypotheses. For example:

Virnstein et al. 1985 [https://doi.org/10.1016/0304-3770\(85\)90021-X](https://doi.org/10.1016/0304-3770(85)90021-X)

Hull 1987 [https://doi.org/10.1016/0272-7714\(87\)90112-0](https://doi.org/10.1016/0272-7714(87)90112-0)

Raffaelli et al 1998 <https://www.taylorfrancis.com/chapters/edit/10.1201/b12646-13/ecological-impact-green-macroalgal-blooms-david-raffaelli-john-raven-lynda-poole>

Norkko et al. 2000 [https://doi.org/10.1016/S0022-0981\(00\)00155-6](https://doi.org/10.1016/S0022-0981(00)00155-6)

Salovius et el 2005 <https://doi.org/10.1016/j.seares.2004.05.001>

Nelson et al 2008 <https://doi.org/10.1890/07-0494.1>

Arroyo et al 2012 <https://doi.org/10.1016/j.jembe.2012.03.020>

Bohórquez et al 2013 <https://doi.org/10.1016/j.marpolbul.2013.02.002>

Arroya et al 2016 <https://doi.org/10.4324/9781315370781-6>

Coffin et al. 2017 <https://doi.org/10.7717/peerj.3080>

Barnes 2019. <https://doi.org/10.1002/aqc.2977>

(Note: this is not an exhaustive list, just the first that came to mind).

On a related note, the introduction also does not provide any motivation for the study, pose any research questions, state hypotheses or expected results. These need to be added

Our response: We highly appreciate the suggestions and the papers listed by the reviewer. The introduction has been edited accordingly and now also includes research questions and states hypotheses/expected results.

Edits in the manuscript:

I. 77-94: “Some previous studies identified a high biodiversity associated with drifting algae in a lagoon off the west coast of the United States³⁰ and in the Baltic Sea³¹⁻³³. Calcifying green algae communities in coral reefs of the Great Barrier Reef, Australia³⁴, and green algae occurring in blooms in the United States³⁵, Canada³⁶ and South Africa³⁷, as well as at further locations of the Atlantic^{38,39}, furthermore, indicate that algal assemblages can support high biodiversity. Additionally, kelp-forming brown algae in the United States⁴⁰ and United Kingdom⁴¹ host a vast array of associated organisms. But, it remains largely unknown whether the mat-forming red alga *P. crispa* that is likely increasing in abundance and potentially replaces classical high biodiversity habitats such as seagrass meadows and rocky substrates harbours high associated sessile biodiversity. Thus, we here aimed to determine the role *P. crispa* as habitat for sessile invertebrate biodiversity. This relates to recent pilot studies that have identified non-colonial²² and sessile polychaetes²⁹ to be associated with *P. crispa* mats. The present study, hence, aims to answer the following research

questions: i) to what extent can *P. crispa* mats function as habitat for sessile invertebrates, and ii) how does this biodiversity compare to neighbouring *Posidonia oceanica* seagrass meadows? Based on aforementioned indications^{22,29}, we hypothesise that *P. crispa* harbours high associated sessile invertebrate diversity. Furthermore, we expect that this biodiversity potentially differs from that of *P. oceanica* due to different habitat characteristics.“

The conclusions of the study are also overstated, also indicating that there needs to be more framing in regards to what is already known and what this study adds to the literature.

In particular, the fact that algal mats can harbour high diversity is not really a surprise – this has been found previously. But previous studies have shown that these algal mats are not great habitat due to their transient and short-lived nature. The discussion touches upon this briefly, but if you are to propose these algal mats as reservoirs of biodiversity, there needs to be more investigation of the actual life span of these algal mats. The potential life span of this algal species is stated at a few decades, but is this true for the study area? It seems that the study deals with drifting algal mats (and I did not find this listed anywhere, so please add) in which case I have a hard time believing that they would actually provide a stable habitat for decades. In reality, they would probably be washed up during storms within a few months to a year.

Our response: We appreciate the points raised by the reviewer. Thus far, *P. crispa* has been suggested to act as habitat for associated invertebrates in the Mediterranean (Bonifazi et al. 2017) and the Black Sea (Zaitsev et al. 2007). To our knowledge, a direct comparison of these *P. crispa* mats with rather classical biodiversity hotspots, such as *P. oceanica* that was selected for the present study, has not been carried out. We agree with the reviewer (and have also added respective information to the manuscript) that algal assemblages could harbour high biodiversity as previously demonstrated for kelp-forming brown algae (Miller et al. 2018, Teagle & Smale 2018) or calcifying green algae (McNeil et al. 2021). In how far mat-forming algae can be considered as biodiversity hotspots remains yet unknown. Findings of the present study extend previous findings (Bonifazi et al. 2017) and provide a direct comparison to neighbouring *P. oceanica* meadows. Based on our results, we conclude that *P. crispa* mats act as temporary reservoirs as the longevity and their persistence is variable. *P. crispa* mats that were sampled for the present study were of attached growth form (i.e., no drifting algae), even though unattached growth forms of *P. crispa* mats have been reported before (Berov et al. 2018). Hence, we respectfully disagree with the reviewer’s point that *P. crispa* mats are washed away within months to a year, particularly due to their presence in depths >10m. In case *P. crispa* mats are dislocated, the manuscript provides two potential scenarios of what could happen to the associated sessile invertebrate biodiversity of *P. crispa* (l. 269-275). Nonetheless we agree with the reviewer that further studies are required to clarify the longevity of *P. crispa* mats in the Mediterranean and included that in the manuscript. The point raised by the reviewer has been added to and discussed within the manuscript. To emphasise the issue of the algae’s longevity, the title of the present manuscript has been edited accordingly.

Edits in the manuscript:

I. 273-286: “We conclude that in both cases, *P. crispa* mats serve as *temporary* ecosystem engineers forming *temporary* refuge habitats, and subsequently as *transitory* biodiversity hotspots. *P. crispa* formations in the Atlantic and Black Sea, however, still provide a relatively stable habitat over several decades²⁵, which underlines the general functioning as a biodiversity substratum. The extent to which this applies to *P. crispa* mats of the Mediterranean as well needs to be determined in future studies. Although dislodged *P. crispa* may not offer a stable environment over

longer time scales compared to the seagrass, the mobile algal thalli may function as an effective dispersal mechanism. The drifting algae themselves may facilitate diverse sessile invertebrate communities as described previously^{30,31}, and could function as a transport vector over relatively large distances³². To which extend the associated phenotypes are tolerant against this drifting behaviour remains speculative³³. Potentially, more tolerant sessile species could reach more favourable areas, e.g., healthy seagrass beds, which are possibly beyond the reach of planktonic larval stages they may exhibit.”

In addition, the study focused mainly on the number of species. However, there is little attention on which species are found in the macroalgae vs the seagrass. Are rare species more common in the seagrass? What about endangered species? What about different taxonomic and functional groups. If the algal mats only support common species and are missing some important taxonomic or trophic or functional groups, they are not effective reservoirs. A species list in the supplement would be helpful as well.

Based on these, toning down the conclusions is needed. Address the limitations of the study and what further information and investigations are needed to determine whether algal mats are truly effective reservoirs in this part of the world. (This applies to the discussion, abstract, and title as well).

Our response: The present study focusses on the number of both phenotype (instead of species, see comment further below) and individuals in *P. crispata* mats and *P. oceanica* meadows. An analysis as intended by the reviewer (e.g., occurrence of e.g., endangered species) was not performed given the fact that a) a taxonomic determination of all taxa to a species level was not possible (see Method section), and b) it goes beyond the scope of the present manuscript. We aimed to quantitatively, and to a certain degree qualitatively, compare the overall sessile invertebrate diversity. An in-depth analysis of specific taxonomic groups was not intended. Furthermore, our biodiversity analysis particularly considers the occurrence of highly frequent, frequent, and rare species (via the concept of Hill numbers). The results part has been edited to include further details on the occurrence of rare species in *P. crispata* vs. *P. oceanica*. Title, abstract and discussion have also been edited based on the suggestions here and further below.

Edit in the manuscript:

I. 1: “Fleshy red algae mats act as temporary reservoirs for sessile invertebrate biodiversity”

I. 29-30: “We here show that the sessile invertebrate biodiversity in these red algae mats is high and exceeds that of neighbouring seagrass meadows.”

I. 32-34: “Our findings suggest that fleshy red algae mats act as alternative habitats and temporary sessile invertebrate biodiversity reservoirs in times of environmental change.”

I. 155-158: “Hence, we could confirm that *P. crispata* harbours assemblages with significantly higher Simpson diversity (~132; Fig 3C, Supplementary Table S3; no overlap of 95% confidence intervals⁴⁶) compared to all other (sub-) habitats, emphasising its role as a biodiversity hotspot for sessile invertebrates.”

I. 178-191: “The difference in estimated phenotype Simpson diversity between *P. crispata* and the *P. oceanica* holobiont at C_{max} is larger than the difference in phenotype richness (i.e., ~50 and ~27, respectively; Supplementary Table S3). Comparing the

empirical richness values (i.e., values for $q = 0$) with the values estimated asymptotically and non-asymptotically (the latter standardised for C_{\max}), it is shown that the number of undetected phenotypes is larger for (sub-) habitats of *P. oceanica* (holobiont and rhizomes) than for *P. crispa* (Supplementary Table S3). These findings indicate that the higher overall diversity in *P. crispa* may be driven by the higher abundance of frequently occurring phenotypes rather than rare phenotypes, while *P. oceanica* seems to harbour a higher number of undetected rare phenotypes. Conclusively, our data allowed us to identify *P. crispa* as a valuable habitat harbouring more even and diverse sessile invertebrate communities that are shaped by the higher abundance of frequently occurring phenotypes compared to neighbouring *P. oceanica* meadows, which in return harbours more undetected rare phenotypes.”

Some more specific comments:

Introduction, general: Somewhere in the introduction or the methods, please add some general ecological information about the study species.

Our response: Information has been added to the manuscript.

Edits in the manuscript:

I. 318-336: “*P. crispa* is a persistent, mat-forming red algae belonging to the order Gigartinales that typically produces branched thalli of up to 15 cm length^{21,25}. It can either grow attached covering hard substrates or unattached growing on sediments²⁶, and it has been observed recently that it grows on reefs engineered by invertebrates²⁶. The red algae used in the present study is a perennial rhodophyte and its mats can be up to 15 cm thick. It tolerates large variations in key environmental parameters (temperature, salinity, light availability)²⁵ and accumulates mostly in depths between 10 and 55 m^{25,26}. *P. crispa* occurs in the Black Sea^{25,26}, the Atlantic^{24,27} and the Mediterranean Sea^{22,23}. The second habitat that was focussed on in the present study are *P. oceanica* meadows that are endemic to the Mediterranean⁷³. Just as meadows constructed by other seagrass species, *P. oceanica* meadows rank amongst the most valuable coastal ecosystems worldwide as they provide a range of goods and ecosystem services^{74,75}, e.g., they exhibit high biodiversity, function as ecosystem engineers, and can act as natural coastal protection barriers⁶⁵. *P. oceanica* meadows consist of the rhizome layer (often up to several m thick)⁶³ and a leaf-canopy part. The meadows occur from shallow waters down to depths of 40 m (depending on water turbidity). Due to anthropogenically-induced environmental stressors⁶, such as nutrient and sediment pollution, habitat loss and degradation¹⁸, pollution^{5,18}, eutrophication^{5,18} and/or ocean warming¹⁸, seagrass meadows range among the most threatened ecosystems worldwide⁷⁶.”

Results and discussion, Lines 81-84: perhaps add the geographic area where this is was studied, so that readers don't have to flip to the methods for this.

Our response: Information has been added to the manuscript.

Edits in the manuscript:

I. 98-100: “We assessed the sessile invertebrate biodiversity in neighbouring *P. crispa* and *P. oceanica* habitats along the north-eastern and north-western coasts of Giglio Island, within the Tuscan Archipelago National park, Tyrrhenian Sea, Italy (see Supplementary Fig. S2).”

Results and discussion, Line 149: elaborate how this is line with recent findings.

Our response: We have added further information as requested. We also discussed in how far the assessed environmental parameters influence observed sessile invertebrate communities.

Edits in the manuscript:

I. 196-227: “Our results show that *P. crispa* shapes key environmental parameters similarly to neighbouring *P. oceanica* seagrass meadows (Supplementary Fig. S1), as particularly water movement and light intensity within the red algae mats as well as in the seagrass meadows are reduced compared to neighbouring bare substrate (Supplementary Fig. S1B, F). This confirms findings of a parallel study that has identified *P. crispa* as an ecosystem engineer modifying its environment²⁸. This function also applies to further environmental parameters: Daily oxygen concentration fluctuations of *P. crispa* (7.73 – 8.14 mg l⁻¹) were similar to those of *P. oceanica* (7.59 – 8.04 mg l⁻¹), with the daily mean of oxygen concentrations being slightly higher in *P. crispa* (7.99 mg l⁻¹) compared to those of *P. oceanica* (7.75 mg l⁻¹). This contrasts previous findings stating that shallow, macroalgae-covered environments undergo wider oxygen concentration fluctuations compared to seagrass meadows^{47,48}. Our findings indicate that this is not necessarily the case in deeper environments (Supplementary Fig. S1). Additionally, pH within *P. crispa* was lower (8.44) compared to *P. oceanica* meadows (8.64). Shaped environmental parameters, particularly reduced water movement and potentially varying oxygen availability, could benefit the settlement of specific bryozoan⁴⁹⁻⁵³, bivalve⁵⁴ and polychaete⁵³ species. This may explain findings of the present study, in which we identified more phenotypes and/or individuals associated with *P. crispa* compared to *P. oceanica* of bryozoans (76 vs. 78 phenotypes, 44222 vs. 7655 indiv. per m² habitat), molluscs (bivalves; 4 vs. 4 phenotypes, 112 vs. 38 indiv. per m² habitat) and polychaetes (23 vs. 13 phenotypes, 5950 vs. 3734)⁵⁵ associated with *P. crispa* compared to *P. oceanica* (see Supplementary Tab. S1). The extent to which further functions such as sediment trapping, related to reduced water movements induced by *P. oceanica* meadows⁵⁶⁻⁵⁸, apply to *P. crispa* mats need to be determined. The complex morphology of *P. crispa* mats, reflected by the surface area enlargement (see next paragraph) and a high surface area (provided by the complexity of the thalli) relative to a small volume (mats of several cm thickness), suggest that sediment and particulate matter are likely trapped. Trapped sediment and particulate matter, hence, could provide a) a heterogeneous habitat for infaunal species⁵⁹ and, b) (in-)organic matter for tube-building species such as sessile polychaetes⁵⁹. Taken together, we conclude that *P. crispa* mats provide a substrate that facilitates the colonisation of sessile organisms^{22,60} by providing (micro-) habitats for associated alpha-diversity (Table 1, Fig. 3), thus, allowing us to propose *P. crispa* as an ecosystem engineer^{1,61}.

Results and discussion, Lines 203-208: References needed here – there are many such references in the literature.

Our response: References have been added.

Edits in the manuscript:

I. 278-286: “Although dislodged *P. crispa* may not offer a stable environment over longer time scales compared to the seagrass, the mobile algal thalli may function as an effective dispersal mechanism. The drifting algae themselves may facilitate diverse sessile invertebrate communities as described previously^{30,31}, and could function as a

transport vector over relatively large distances³². To which extend the associated phenotypes are tolerant against this drifting behaviour remains speculative³³. Potentially, more tolerant sessile species could reach more favourable areas, e.g., healthy seagrass beds, which are possibly beyond the reach of planktonic larval stages they may exhibit.“

Methods: what the red algae sampled attached or drifting? This is likely an important distinction in terms of the longevity and stability of the habitat.

Our response: The *P. crispa* mats sampled for the present study were growing attached to rocky substrates. This information has been added to the manuscript.

Edits in the manuscript:

I. 342-343: “An attached growth form of *P. crispa* was chosen for the present study.”

Methods: The methods are mostly very detailed and the code is available, so that is excellent! I’m not sure about the use of the word phenotypes throughout the manuscript. I understand why it was chosen, but think the word “species” or “taxa” could replace it (with a note in the methods noting that several species might be represented separately). Otherwise, the word phenotype would be more accurate to represent different types of individuals within a single species, rather than many.

Our response: We acknowledge the point raised by the reviewer. However, we respectfully disagree and refrained from replacing the word “phenotype” for two reasons: a) we were not able to fully identify all taxa to a species-level (as stated in the Material section). By using the words species throughout the manuscript and solely giving a note in the Material-section about this would be inaccurate and would then, b) contribute to the “overselling”-aspect that the reviewer has raised earlier.

Tables: Please add a species list in the supplement that shows the species found in the different habitats.

Our response: We have added a species list that includes information about the determination level as well as about the occurrence in the respective habitat. The list is accessible via El-Khaled et al. (2021)⁴⁶ (see data availability statement).

Figures: Please change the colour palette! The red and green colours used are completely indistinguishable to anybody who is colour-blind. This includes the following figures: Figure 2, Figure 3, and Figure S1

Our response: We appreciate the point raised by the reviewer and support the barrier-free access to science. We have changed the colour palette of the mentioned figures accordingly.

Edits in the manuscript:

Fig. 2:

Fig. 3:

Fig. S1:

Reviewers' comments:

Reviewer #1 (Remarks to the Author):

I thank the authors for their revisions to the first version of this manuscript. Most of my comments were satisfactorily addressed. However, a handful were not, and the re-review sparked some additional minor suggestions that I hope the authors will consider for the next version.

Justification for sessile invertebrate epifauna over other groups – The authors' response to my comment 2 is informative but not reflected in the revised text, except for the first point involving pilot studies. I do not think it negates the value of the study to admit that only sessile invertebrates were assessed because there was readily available prior data on sessile invertebrates, because mobile invertebrates could not be adequately sampled with the chosen methodology, and because sessile invertebrates were of interest because they mean the habitat is stable for more than a few days (or longer, depending on species). But such justifications should all be included in the text.

Ecosystem engineering functions – This section of the discussion has been improved. However, on reflection, I suggest two further modifications. First, because the authors present influence on water quality parameters as a key argument for the ecological value of *P. crispata* mats, I think the environmental data figure should be brought into the main text rather than relegated to the supplemental material. Ranges, daily means, percentage differences, or other descriptions of all the parameters should be presented and their potential relevance discussed. This is currently only done for oxygen concentrations. If the authors do move the figure to the main text, I also suggest some modifications to the figure. Slightly thicker lines and a different color scheme would improve visibility (e.g. the same purple/green/dark yellow color scheme used for the other main figures). The y-axis and caption for panel F should be updated to indicate that it is weight loss of a clod card, and an explanation of the statistics used in that panel should be presented.

Second, because the water quality data are from a short time frame from one site (which is understandable, given the logistics involved), I think the authors could consider slightly softening some language in this section to reflect the limitations of drawing conclusions from spatiotemporally limited data. For example, "suggest" could be used in certain places instead of "show" or "indicate". As another example, the statement that the data "confirm" those in Schmidt et al. is a bit misleading if both datasets were collected during the same two months at the same general site.

Writing clarity – There is much improvement from the first version, but I still think a style edit would be beneficial. The ways in which many sentences are broken by modifiers (e.g. thus, hence, nonetheless, subsequently) and reference other places in the text (e.g. see above, see below, aforementioned) slowed my reading of both versions of this manuscript considerably. Syntax that I found confusing in the first version was not fixed (e.g. comment 22, now p. 12 L. 267-272; comment 30, now p. 19 L. 426-428). There are a few new grammatical issues (e.g. p. 14 L. 305-308). If these will be addressed at the proofreading stage, then this comment can be ignored.

p. 5 L. 89-94: The inclusion of an expected result in response to the other reviewer's comments is a plus. However, the expectation is very vaguely worded. "Differs" does not tell us the direction of the expected difference (higher? lower?), and "different habitat characteristics" could be more specific (which characteristics did the authors expect to be influential?)

p. 6 L. 104: Perhaps "covering 9 higher taxa" or "covering 9 taxa at subclass level or above" or just omit this phrase? Order, family, genus, species, etc. are also taxa, but I assume the authors are referring to the 9 groups listed in the methods section.

p. 21 L. 479-497: This comment was not fully addressed. PERMANOVA is still misspelled twice. The rationale for using both PERMANOVA and ANOSIM is still not given. They do test slightly different hypotheses but seem to be used for the same redundant purpose here (to show that sessile invertebrate communities are distinct among *P. crispata*, *P. oceanica* leaves, and *P. oceanica*

rhizomes).

Figure 2 caption: Please make sure the figure description reflects the new color scheme used in the figure.

Methods section: Some of the information on *P. crispa* and biodiversity calculations that was moved earlier in the manuscript according to the reviewers' recommendations remains word-for-word in the methods section.

Reviewer #2 (Remarks to the Author):

First of all, I would like to congratulate the authors on this revised manuscript – it's clear that there has been a lot of hard work and good consideration of reviewer comments. The new figures and colour scheme also look great! I do still have a few comments/revisions that would need to be addressed. Most of these are minor, style and polishing issues, but I believe the Results/Discussion section could still be improved and in particular, shortened to improve readability and avoid repetition.

Lines 70-75: please rephrase these sentences so they don't all start with "P. crispa" – it will be easier to read.

Lines 77-78: Please insert a paragraph break here.

Line 94: Explain here or earlier which habitat characteristics differ.

Line 104: I still disagree with the use of the word phenotype here, though I understand why its used. Anyway, if you do ahead with the use of phenotype, please add an explanation of why you are using here. (similar to what you wrote in the rebuttal).

Lines 116, 117, 118, and elsewhere: Please use a space as a thousand separator instead of a comma, to avoid confusion with decimal points.

Lines 213/214 and 216: The phrase "associated with *P.crispa* compared with *P. oceanica*" is repeated here – just delete one instance.

General: Overall, the results/discussion is improved, but I would suggest a thorough round of editing. There is quite a bit of repetition, and similar concepts brought up multiple times in different places, and it could be shortened considerably. For example, you could merge 219-227 with the next paragraph and remove a lot of repetition. Similarly, lines 279-286 could be moved up near lines 269-272. These are just examples, but some critical editing is needed to tighten it up and make a clearer "storyline".

Lines 254 and 300: Here, you point out the "considerable overlap" of shared phenotypes. However, you contradict this in other places (e.g. lines 106-107: a high number of unique phenotypes, line 304: "the clear distinction of sessile invertebrate communities"). From Fig 2A, it seems that there is not that much overlap: ~half of the phenotypes in *Posidonia* are not found in *P. crispa*. Therefore you need to give this more consideration when proposing *P. crispa* as a "backup reservoir" and explicitly state for "some species" not in general, and that alone it cannot conserve biodiversity.

Reviewer #1 (Remarks to the Author):

I thank the authors for their revisions to the first version of this manuscript. Most of my comments were satisfactorily addressed. However, a handful were not, and the re-review sparked some additional minor suggestions that I hope the authors will consider for the next version.

Our response: We would like to thank the reviewer for the extremely constructive and helpful feedback. Please find the point-to-point response targeting the additional suggestions.

Justification for sessile invertebrate epifauna over other groups – The authors' response to my comment 2 is informative but not reflected in the revised text, except for the first point involving pilot studies. I do not think it negates the value of the study to admit that only sessile invertebrates were assessed because there was readily available prior data on sessile invertebrates, because mobile invertebrates could not be adequately sampled with the chosen methodology, and because sessile invertebrates were of interest because they mean the habitat is stable for more than a few days (or longer, depending on species). But such justifications should all be included in the text.

Our response: Further justifications have been added to the introduction.

Edit in the manuscript:

I. 85-102: "It remains unknown whether the mat-forming red alga *P. crispera*, which increases in abundance and potentially replaces classical high biodiversity habitats also harbours high associated sessile biodiversity. Based on recent pilot studies that have identified non-colonial²² and sessile polychaetes²⁹ to be associated with *P. crispera* mats, we here determined the role of *P. crispera* as habitat for overall sessile invertebrate biodiversity. The present study aims to answer the following research questions: i) to what extent can *P. crispera* mats function as habitat for sessile invertebrates, and ii) how does this biodiversity compare to neighbouring *Posidonia oceanica* seagrass meadows? We focussed on the sessile biodiversity in *P. crispera* mats and adjacent *P. oceanica* meadows for several reasons. Firstly, previous pilot studies^{22,29} lead to the hypothesis of a high associated sessile invertebrate diversity in *P. crispera* that is comparable in terms of community composition, invertebrate species richness and abundance to that of *P. oceanica*. This invertebrate diversity is likely linked to different habitat characteristics such as micro-niches evoked by varying influences on key environmental parameters and ecosystem engineering functions²⁸. Secondly, a particular sampling procedure was chosen to ensure the complete retrieval of sessile invertebrates causing a potential loss of mobile invertebrates. Thirdly, the presence of sessile invertebrates potentially reflects the stability and longevity of the red algae mats as habitats⁴²."

Ecosystem engineering functions – This section of the discussion has been improved. However, on reflection, I suggest two further modifications. First, because the authors present influence on water quality parameters as a key argument for the ecological value of *P. crispera* mats, I think the environmental data figure should be brought into the main text rather than relegated to the supplemental material. Ranges, daily means, percentage differences, or other descriptions of all the parameters should be presented and their potential relevance discussed. This is currently only done for oxygen concentrations. If the authors do move the figure to the main text, I also suggest some modifications to the figure. Slightly thicker lines and a different color scheme would improve visibility

(e.g. the same purple/green/dark yellow color scheme used for the other main figures). The y-axis and caption for panel F should be updated to indicate that it is weight loss of a clod card, and an explanation of the statistics used in that panel should be presented.

Our response: We agree with the reviewer that moving the respective figure from the supplementary to the main body of the manuscript is an asset. The lines have been slightly thickened, as well as a colour scheme analogue to Fig. 3 (purple and blue for *P. crispa* and *P. oceanica*, resp.) has been chosen. Edits on panel F have been performed. Furthermore, additional information about the environmental parameters (exceeding oxygen concentrations) has been added to the manuscript as suggested.

Edit in the manuscript:

I. 217-228: “Additionally, the average pH within *P. crispa* mats was lower (8.44) compared to *P. oceanica* meadows (8.64), which resembled the observed differences in O₂ concentrations (Fig. 4A, C). Photosynthesis by algae and plants requires hydrogen ions, which results in increased pH levels while respiration lowers pH levels^{50,51}. Furthermore, our data suggest higher light availability in *P. crispa* (538 lux) compared to *P. oceanica* meadows (315 lux; Fig. 4B) at the same depth. These findings corroborate with previous studies that identified strong light attenuations in macrophyte habitats due to self-shading effects^{52,53}. A lessened self-shading effect in the red algae habitat compared to the *P. oceanica* seagrass meadows could be explained by the comparatively thin morphology of the algae mats. Finally, the reduced water movement (Fig. 4F) and varying O₂ availability (Fig. 4A) may benefit the settlement of specific bryozoan⁵⁴⁻⁵⁸, bivalve⁵⁹ and polychaete⁵⁸ species.”

I. 233-246: “The higher number of phenotypes and individuals in *P. crispa* relative to *P. oceanica* may be partly explained by the specific surface area that potentially offers substrate, and thus micro-habitats for mobile and sessile invertebrates^{22,25}. The complex morphology of *P. crispa* mats is reflected in the 2D to 3D surface area enlargement factor. Here, a high surface area provided by complex thalli relative to a small volume (mats of several cm thickness) resulted in an enlargement factor of 4.9 ± 0.2 (mean \pm standard error; Supplementary Table S2) for *P. crispa*, which is lower than for *P. oceanica* (both leaves (7.3 ± 0.5) and the *P. oceanica* holobiont (8.3 ± 0.5) but higher than for *P. oceanica* rhizomes (2.0 ± 0.1)). This structural complexity may also explain the observed reduced water movement within *P. crispa* mats (Fig. 4F) that could favour sediment trapping. The extent to which further functions such as sediment trapping, similar to the reduced water movements induced by *P. oceanica* meadows⁶¹⁻⁶³, apply to *P. crispa* mats need to be determined in future studies. Trapped sediment and particulate matter could provide 1) a heterogeneous habitat for infaunal species⁶⁴ and, 2) (in-) organic matter for tube-building species such as sessile polychaetes⁶⁴.”

I. 910-915: “**Figure 4: Environmental data** consisting of oxygen (O₂) concentration (A), light intensity (B), pH (C), temperature (D), chlorophyll *a* concentration (E), and water movement (estimated via weight loss of clod cards; F) in *Phyllophora crispa* (purple), *Posidonia oceanica* (blue) and neighbouring hard bottom substrate serving as a reference habitat (brown). Horizontal lines within panels A-E display daily mean of respective parameters in each habitat. Note for panel F: different letters above box plots indicate significant differences between habitats (ANOVA and subsequent Tukey HSD test).”

Second, because the water quality data are from a short time frame from one site (which is understandable, given the logistics involved), I think the authors could consider slightly softening some language in this section to reflect the limitations of drawing conclusions from spatiotemporally limited data. For example, “suggest” could be used in certain places instead of “show” or “indicate”. As another example, the statement that the data “confirm” those in Schmidt et al. is a bit misleading if both datasets were collected during the same two months at the same general site.

Our response: The section was toned down according to the reviewers’ suggestions.

Edit in the manuscript:

I. 205-207: “. Our results suggest that *P. crispata* shapes key environmental parameters similarly to neighbouring *P. oceanica* seagrass meadows (Fig. 4).”

I. 209-210: “This extends findings of a parallel study that has identified *P. crispata* as an ecosystem engineer modifying its environment²⁸.”

I. 210-211: “This functioning as an ecosystem engineer seems to apply to further environmental parameters.”

I. 254-256: “We conclude that *P. crispata* mats facilitate the colonisation of sessile organisms^{22,65} by providing (micro-) habitats for associated alpha-diversity (Table 1, Fig. 3), thus, allowing us to propose *P. crispata* as an ecosystem engineer^{1,66}.”

Writing clarity – There is much improvement from the first version, but I still think a style edit would be beneficial. The ways in which many sentences are broken by modifiers (e.g. thus, hence, nonetheless, subsequently) and reference other places in the text (e.g. see above, see below, aforementioned) slowed my reading of both versions of this manuscript considerably. Syntax that I found confusing in the first version was not fixed (e.g. comment 22, now p. 12 L. 267-272; comment 30, now p. 19 L. 426-428). There are a few new grammatical issues (e.g. p. 14 L. 305-308). If these will be addressed at the proofreading stage, then this comment can be ignored.

Our response: We have focused on the writing clarity and edited the manuscript accordingly.

Edits in the manuscript (besides others):

I. 54-58: “Identifying potential biodiversity refugia pivotal for rebuilding marine life¹⁹ is therefore needed to appropriately adapt conservation strategies in times of increased biodiversity loss associated with anthropogenic global change^{11,12,17} and direct local human impacts (e.g., pollution, coastal development)^{5,6,18}.”

I. 270-276: “Even though *P. crispata* mats harbour sessile invertebrates in numbers that exceed those of neighbouring *P. oceanica* meadows⁶⁰ and other ecosystems (Table 1), these mats substantially differ from seagrass meadows in their longevity. In the Mediterranean, *P. oceanica* meadows form dense rhizome layers which can be of several meter thickness when admixed with trapped sediment⁶⁸. Similar to coral reefs or mangrove forests, seagrass meadows can persist several millennia⁶⁹, which exceeds the current estimated lifespan of *P. crispata* formations (i.e., decades)²⁵.”

I. 281-283: “In contrast to *P. oceanica* meadows, *P. crispata* mats can be dislodged and translocated by waves²⁵, particularly those with an unattached growth form on sediments²⁶.”

I. 317-320: “Our findings suggest that *P. crispata* mats and their associated sessile invertebrate communities potentially aid reviving classical marine (sessile invertebrate) biodiversity hotspots such as invaluable seagrass meadows in the Mediterranean Sea once threats are reduced or removed^{19,74,76}.”

I. 429-431: “We performed the described workflow according to the following steps to achieve meaningful comparisons of investigated biotic communities (see Chao et al. (2020)⁴⁶ and Supplementary Table S3):“

p. 5 L. 89-94: The inclusion of an expected result in response to the other reviewer’s comments is a plus. However, the expectation is very vaguely worded. “Differs” does not tell us the direction of the expected difference (higher? lower?), and “different habitat characteristics” could be more specific (which characteristics did the authors expect to be influential?)

Our response: Has been specified.

Edit in the manuscript:

I. 92-102: “We focussed on the sessile biodiversity in *P. crispa* mats and adjacent *P. oceanica* meadows for multiple reasons. Firstly, previous pilot studies^{22,29} lead to the hypothesis of a high associated sessile invertebrate diversity in *P. crispa* that is comparable in terms of community composition, invertebrate species richness and abundance to that of *P. oceanica*. This invertebrate diversity is likely linked to different habitat characteristics such as micro-niches evoked by varying influences on key environmental parameters and ecosystem engineering functions²⁸. Secondly, a particular sampling procedure was chosen to ensure the complete retrieval of sessile invertebrates causing a potential loss of mobile invertebrates. Thirdly, the presence of sessile invertebrates potentially reflects the stability and longevity of the red algae mats as habitats⁴².”

p. 6 L. 104: Perhaps “covering 9 higher taxa” or “covering 9 taxa at subclass level or above” or just omit this phrase? Order, family, genus, species, etc. are also taxa, but I assume the authors are referring to the 9 groups listed in the methods section.

Our response: Has been edited accordingly.

Edit in the manuscript:

I. 115-117: “We recorded 312 distinct sessile invertebrate phenotypes (covering 9 higher taxa) for both *P. crispa* and *P. oceanica*, of which 223 occurred in *P. crispa* mats and 179 in *P. oceanica* holobionts, respectively (Fig. 2A).”

p. 21 L. 479-497: This comment was not fully addressed. PERMANOVA is still misspelled twice. The rationale for using both PERMANOVA and ANOSIM is still not given. They do test slightly different hypotheses but seem to be used for the same redundant purpose here (to show that sessile invertebrate communities are distinct among *P. crispa*, *P. oceanica* leaves, and *P. oceanica* rhizomes).

Our response: PERMANOVA is now spelled correctly. We agree to the reviewer and therefore excluded the ANOSIM analysis from the manuscript.

Figure 2 caption: Please make sure the figure description reflects the new color scheme used in the figure.

Our response: Has been checked and edited.

Methods section: Some of the information on *P. crispa* and biodiversity calculations that was moved earlier in the manuscript according to the reviewers’ recommendations remains word-for-word in

the methods section.

Our response: Ecological information about *P. crispa* in the method section has been deleted to avoid repetitiveness as it is present in the main part of the manuscript.

Reviewer #2 (Remarks to the Author):

First of all, I would like to congratulate the authors on this revised manuscript – it's clear that there has been a lot of hard work and good consideration of reviewer comments. The new figures and colour scheme also look great! I do still have a few comments/revisions that would need to be addressed. Most of these are minor, style and polishing issues, but I believe the Results/Discussion section could still be improved and in particular, shortened to improve readability and avoid repetition.

Our response: We highly appreciate the previous and current comments that improved the present manuscript. Please find detailed information below.

Lines 70-75: please rephrase these sentences so they don't all start with "P. crispera" – it will be easier to read.

Our response: Sentences have been rephrased accordingly.

Edit in the manuscript:

I. 71-78: "*P. crispera* is a perennial rhodophyte of the order Gigartinales that typically produces branched thalli of up to 15 cm length^{21,25}. These red algae mats tolerate large variations in key environmental parameters and can proliferate under low water temperature (< 10 °C) and salinity (18 PSU)²⁵. *P. crispera* is sciaphilic^{22,26}, i.e., adapted to low-light conditions, and reaches large accumulations in water depths between 10 and 55 m^{25,26}. Its thalli can exhibit an either attached growth form covering hard substrates or an unattached form growing on sediments²⁶, with recent observations from the Black Sea displaying that *P. crispera* also grows on reefs engineered by invertebrates²⁶."

Lines 77-78: Please insert a paragraph break here.

Our response: Has been inserted.

Line 94: Explain here or earlier which habitat characteristics differ.

Our response: Additional information has been added.

Edit in the manuscript:

I. 92-102: "We focussed on the sessile biodiversity in *P. crispera* mats and adjacent *P. oceanica* meadows for several reasons. Firstly, previous pilot studies^{22,29} lead to the hypothesis of a high associated sessile invertebrate diversity in *P. crispera* that is comparable in terms of community composition, invertebrate species richness and abundance to that of *P. oceanica*. This invertebrate diversity is likely linked to different habitat characteristics such as micro-niches evoked by varying influences on key environmental parameters and ecosystem engineering functions²⁸. Secondly, a particular sampling procedure was chosen to ensure the complete retrieval of sessile invertebrates causing a potential loss of mobile invertebrates. Thirdly, the presence of sessile invertebrates potentially reflects the stability and longevity of the red algae mats as habitats⁴²."

Line 104: I still disagree with the use of the word phenotype here, though I understand why its used. Anyway, if you do ahead with the use of phenotype, please add an explanation of why you are using

here. (similar to what you wrote in the rebuttal).

Our response: A definition of “phenotype” has been added to the main text.

Edit in the manuscript:

I. 111-114: “Briefly, invertebrates were determined to the lowest possible taxonomic level. However, in case no clear identification was possible, individuals were distinguished based on distinct visual characteristics, resulting in the identification of distinct phenotypes rather than species.”

Lines 116, 117, 118, and elsewhere: Please use a space as a thousand separator instead of a comma, to avoid confusion with decimal points.

Our response: Has been edited throughout the manuscript.

Lines 213/214 and 216: The phrase “associated with *P. crispera* compared with *P. oceanica*” is repeated here – just delete one instance.

Our response: Has been edited according to the reviewers’ suggestion.

Edit in the manuscript:

I. 228-232: “This may explain findings of the present study, in which we identified more phenotypes and/or individuals associated with *P. crispera* compared to *P. oceanica* of bryozoans (76 vs. 78 phenotypes, 44 222 vs. 7 655 indiv. per m² habitat), molluscs (bivalves; 4 vs. 4 phenotypes, 112 vs. 38 indiv. per m² habitat) and polychaetes (23 vs. 13 phenotypes, 5 950 vs. 3 734 indiv. per m² habitat⁶⁰; see Supplementary Tab. S1).”

General: Overall, the results/discussion is improved, but I would suggest a thorough round of editing. There is quite a bit of repetition, and similar concepts brought up multiple times in different places, and it could be shortened considerably. For example, you could merge 219-227 with the next paragraph and remove a lot of repetition. Similarly, lines 279-286 could be moved up near lines 269-272. These are just examples, but some critical editing is needed to tighten it up and make a clearer “storyline”.

Our response: We have edited the manuscript and adjusted the given examples.

Edit in the manuscript (besides others):

I. 233-260: “The higher number of phenotypes and individuals in *P. crispera* relative to *P. oceanica* may be partly explained by the specific surface area that potentially offers substrate, and thus micro-habitats for mobile and sessile invertebrates^{22,25}. The complex morphology of *P. crispera* mats is reflected in the 2D to 3D surface area enlargement factor. Here, a high surface area provided by complex thalli relative to a small volume (mats of several cm thickness) resulted in an enlargement factor of 4.9 ± 0.2 (mean \pm standard error; Supplementary Table S2) for *P. crispera*, which is lower than for *P. oceanica* (both leaves (7.3 ± 0.5) and the *P. oceanica* holobiont (8.3 ± 0.5) but higher than for *P. oceanica* rhizomes (2.0 ± 0.1)). This structural complexity may also explain the observed reduced water movement within *P. crispera* mats (Fig. 4F) that could favour sediment trapping. The extent to which further functions such as sediment trapping, similar to the reduced water movements induced by *P. oceanica* meadows⁶¹⁻⁶³, apply to *P. crispera* mats need to be determined in future studies. Trapped sediment and particulate matter could provide 1) a heterogeneous habitat for infaunal species⁶⁴ and, 2) (in-) organic matter for tube-building species such as sessile polychaetes⁶⁴. Growth form, enlargement factor and persistence²⁵ of *P. crispera* contradict the common notion that structural complexity is reduced when spatially

complex and long-living habitats, such as seagrass meadows, decline^{2,6}. We further estimated the number of individuals per area m² of seafloor by multiplying the calculated numbers of individuals per habitat m² with the respective enlargement factor (Supplementary Table S2). *P. crispata* supported 313 635 ± 27 486 individuals per seafloor m², which is approximately twice that of the *P. oceanica* holobiont (162 139 ± 11 794 individuals per m²; Dunn's test p = 0.009).

We conclude that *P. crispata* mats facilitate the colonisation of sessile organisms^{22,65} by providing (micro-) habitats for associated alpha-diversity (Table 1, Fig. 3), thus, allowing us to propose *P. crispata* as an ecosystem engineer^{1,66}. Together with the considerable surface area enlargement (Supplementary Table S2), environmental parameters shaped by *P. crispata* (Fig. 4), its wide distribution^{22,24,26,27} and the comparative biodiversity analysis (Table 1, Fig. 2, 3), red algae mats may function as overlooked ecosystem engineers and harbour high sessile invertebrate biodiversity.“

I. 281-299: “In contrast to *P. oceanica* meadows, *P. crispata* mats can be dislodged and translocated by waves²⁵, particularly those with an unattached growth form on sediments²⁶. Although dislodged *P. crispata* may not offer a stable environment over longer time scales, mobile algal thalli may function as an effective dispersal mechanism. Drifting algae parts may offer substrate to diverse sessile invertebrate communities^{30,31}, and function as a transport vector over large distances³². The extent to which the associated phenotypes identified in this study tolerate this drifting behaviour remains speculative³³. The translocation of *P. crispata* mats may have consequences for associated biodiversity through two pathways: i) translocated *P. crispata*²⁵, which can colonise and spread vegetative, may still provide habitat for associated sessile invertebrates; or ii) *P. crispata* mats are severely damaged, losing their function as ecosystem engineers, and, hence, biodiversity hotspots. We conclude that in both cases, *P. crispata* mats serve as *temporary* ecosystem engineers forming *temporary* refuge habitats, and subsequently as *transitory* biodiversity hotspots. Potentially, more tolerant sessile species could reach more favourable areas such as healthy seagrass beds that are possibly beyond the reach of planktonic larval stages. *P. crispata* formations in the Atlantic and Black Sea provide a relatively stable habitat over several decades²⁵, which underlines the general functioning as a biodiversity substratum. The extent to which this function applies to *P. crispata* mats of the Mediterranean as well needs to be determined in future studies.”

Lines 254 and 300: Here, you point out the “considerable overlap” of shared phenotypes. However, you contradict this in other places (e.g. lines 106-107: a high number of unique phenotypes, line 304: “the clear distinction of sessile invertebrate communities”). From Fig 2A, it seems that there is not that much overlap: ~half of the phenotypes in *Posidonia* are not found in *P. crispata*. Therefore you need to give this more consideration when proposing *P. crispata* as a “backup reservoir” and explicitly state for “some species” not in general, and that alone it cannot conserve biodiversity.

Our response: Has been toned down and edited accordingly.

Edit in the manuscript:

I. 266-269: “The high biodiversity associated with red algae *P. crispata* mats may positively impact sessile invertebrate communities in bordering *P. oceanica* seagrass meadows⁶⁷, which is reflected by a total of 90 shared phenotypes that occur in all investigated habitats (Fig. 2A).”

I. 308-320: “In the Mediterranean, we hypothesise that *P. crispata* can support *P. oceanica* meadows (and other habitats, see Table 1) by maintaining their sessile

invertebrate biodiversity^{67,77}, particularly due to an overlap of shared phenotypes, i.e., sessile invertebrates that occur in both *P. crispata* and *P. oceanica* (Fig. 2A), even though both habitats harbour a range of unique phenotypes. It remains to be determined i) to what extent the community composition in re-colonised *P. oceanica* meadows differs from their initial sessile invertebrate community composition, considering the clear distinction of associated sessile invertebrate communities in *P. crispata* mats and *P. oceanica* meadows (Fig. 2B), and ii) whether this function applies to all shared phenotypes and potentially further taxa. Our findings suggest that *P. crispata* mats and their associated sessile invertebrate communities potentially aid reviving classical marine (sessile invertebrate) biodiversity hotspots such as invaluable seagrass meadows in the Mediterranean Sea once threats are reduced or removed^{19,74,76} „

Reviewers' comments:

Reviewer #1 (Remarks to the Author):

I thank the authors for their revisions. Some minor comments are below.

The only major comment I have is that I still believe a full style edit needs to be performed. The other reviewer and I have both commented multiple times on this subject, and the authors have made improvements during each round of revision. However, I find that the syntax and flow of ideas remain convoluted in many places in the manuscript. Truly addressing this might be a task for a professional editor. But because of that, I defer to the opinion of the other reviewer and to the managing editor as to whether they also feel further language revision is necessary.

- In the Methods, please mention somewhere what p-values were considered significant.
- Somewhere (I think the supplementary material would be fine), the full results of the PERMANOVA, test of multivariate dispersion, Kruskal-Wallis tests, and Dunn's tests would be helpful. Only p-values or no results are given for these tests in the Results.
- L. 194-196 and 196-200 are nearly redundant sentences (also, do you think that the potentially lower number of undetected rare phenotypes in *P. crispa* mats lessens their value as a refuge habitat?)
- L. 223 – Both seagrasses and macroalgae are macrophytes.
- L. 225-226 – What is thin morphology? In what dimension? Thickness of the mat? Opacity of the blades? Density of thalli?
- L. 226-227 – Reduced water movement and varying O₂ availability in which habitats? *Posidonia* compared to *P. crispa*, or both compared to hard substrate? Since you say "specific" species, are there any species of bryozoan, bivalve, or polychaete in the references cited that are of particular ecological importance that the reader should be aware of? Also, do you think lower pH in *P. crispa* mats might cancel other potential benefits like higher average O₂ levels for organisms like recently settled bivalves?
- L. 331-339 – This paragraph probably belongs in the Introduction.
- Figure 4 – Please note in the caption whether these are average daily cycles (for how many samples) or a representative daily cycle. The number of samples in Panel F should also be either in Panel F or in the caption.

Reviewer #2 (Remarks to the Author):

Thank you for revisions. I'm satisfied with the work done here, and have just a few minor additional corrections to note:

Lines 59-62: These two sentences are essentially stating the same things – can they be combined?

Line 83: "in the Atlantic"

Lines 99-102: I would remove the second reason here and mention it after the others, not as a reason, but as a consequence of choosing to focus on sessile fauna.

Line 125 and throughout the manuscript: change "indiv. per m²" to "ind m⁻²"

Reviewer #1 (Remarks to the Author):

I thank the authors for their revisions. Some minor comments are below.

Our response: We thank the reviewer for the helpful and constructive feedback. Please find the point-to-point response targeting the suggestions below.

The only major comment I have is that I still believe a full style edit needs to be performed. The other reviewer and I have both commented multiple times on this subject, and the authors have made improvements during each round of revision. However, I find that the syntax and flow of ideas remain convoluted in many places in the manuscript. Truly addressing this might be a task for a professional editor. But because of that, I defer to the opinion of the other reviewer and to the managing editor as to whether they also feel further language revision is necessary.

Our response: We have addressed this by editing the manuscript thoroughly regarding its style, and particularly regarding syntax and flow of ideas. Additionally, a native British English speaker has performed an extensive editing of our manuscript. Her edits have been used to further improve the manuscript.

- In the Methods, please mention somewhere what p-values were considered significant.

Our response: This information has been added to the method section.

Edits in the manuscript:

I. 484-487: “Non-parametric permutational multivariate analysis of variance (PERMANOVA⁹⁷; based on species abundance data using Primer-E v6⁹⁸ with the PERMANOVA+ extension)⁹⁹ was used to check for significant differences (i.e., $p \leq 0.05$) in the sessile invertebrate community composition among (sub-) habitats.”

- Somewhere (I think the supplementary material would be fine), the full results of the PERMANOVA, test of multivariate dispersion, Kruskal-Wallis tests, and Dunn’s tests would be helpful. Only p-values or no results are given for these tests in the Results.

Our response: Requested information has been added to the supplementary material (Supplementary Tables S2-S4).

- L. 194-196 and 196-200 are nearly redundant sentences (also, do you think that the potentially lower number of undetected rare phenotypes in *P. crispa* mats lessens their value as a refuge habitat?)

Our response: We have adjusted the sentences in line 194-200 as suggested. A rationale for the suggesting *P. crispa* as a valuable refuge habitat has been added.

Edits in the manuscript:

I. 199-206: “These findings indicate that the higher overall diversity in *P. crispa* may be driven by the higher abundance of frequently occurring rather than rare phenotypes. However, even though the estimated number of undetected phenotypes was higher for *P. oceanica* compared to *P. crispa*, the overall estimated diversity for all orders of *q* in the red algae habitats still remained higher relative to the seagrass meadows (Fig. 2C). Taken together, our data have identified *P. crispa* as a habitat that harbours more even and diverse sessile invertebrate communities compared to neighbouring *P. oceanica* meadows.”

- L. 223 – Both seagrasses and macroalgae are macrophytes.

Our response: Additional information has been added to avoid misunderstandings.

Edits in the manuscript:

I. 228-230: “These findings corroborate with previous studies that identified strong light attenuations in seagrass macrophyte habitats due to self-shading effects^{57,58}.”

- L. 225-226 – What is thin morphology? In what dimension? Thickness of the mat? Opacity of the blades? Density of thalli?

Our response: Has been clarified.

Edits in the manuscript:

I. 230-233: “A lessened self-shading effect in the red algae habitat compared to the *P. oceanica* seagrass meadows could be explained by morphological differences between the two habitats. The latter forms meadows of higher thickness relative to the mats formed by *P. crispa*, with *P. oceanica* leaves being wider than thalli of *P. crispa*.”

- L. 226-227 – Reduced water movement and varying O₂ availability in which habitats? *Posidonia* compared to *P. crispa*, or both compared to hard substrate? Since you say “specific” species, are there any species of bryozoan, bivalve, or polychaete in the references cited that are of particular ecological importance that the reader should be aware of? Also, do you think lower pH in *P. crispa* mats might cancel other potential benefits like higher average O₂ levels for organisms like recently settled bivalves?

Our response: Has been clarified. We added information from cited references to specific species that may have been detected in our study. Additional information regarding the potential effects of lower pH on settled bivalves has been added to the manuscript.

Edits in the manuscript:

I. 233-236: “Finally, the reduced water movement (Fig. 4F) in both habitats and higher O₂ availability in *P. crispa* compared to *P. oceanica* (Fig. 4A) may benefit the settlement of

specific bryozoans (e.g., *Bugula* sp., *Schizoporella* sp.)^{59–63}, bivalves⁶⁴ and polychaetes (*Hydroides* sp.)⁶³.”

I. 241-245: “Potentially, lower pH in *P. crispera* mats (Fig. 4C) may have limited the presence of organisms such as bivalves⁶⁶ or benefitted comparatively resilient organisms such as specific bryozoans⁶⁷. Hence, the extent to which lower pH conditions in *P. crispera* compared to *P. oceanica* may have altered potential benefits such as higher O₂ availability (Fig. 4A) remains speculative.”

- L. 331-339 – This paragraph probably belongs in the Introduction.

Our response: The paragraph has been integrated into the introduction as suggested by the reviewer.

Edits in the manuscript:

I. 59-69: “In the Mediterranean Sea, rocky hard-bottom communities and commonly identified biodiversity hotspots such as seagrass meadows are declining primarily due to environmental pressures^{5,6,19,20}. Meadows formed by *Posidonia oceanica* seagrass rank amongst the most valuable coastal ecosystems worldwide as they provide a range of goods and ecosystem services^{21,22}, e.g., they exhibit high biodiversity, function as ecosystem engineers, and can act as natural coastal protection barriers²³. *P. oceanica* meadows consist of the rhizome layer (often up to several m thick)²⁴ and the leaf-canopy. The meadows occur from shallow waters down to depths of 40 m (depending on water turbidity). Due to anthropogenically-induced environmental stressors⁶, such as nutrient and sediment pollution, habitat loss and degradation¹⁹, pollution^{5,19}, eutrophication^{5,19} and/or ocean warming¹⁹, seagrass meadows are among the most threatened ecosystems worldwide²⁵.”

- Figure 4 – Please note in the caption whether these are average daily cycles (for how many samples) or a representative daily cycle. The number of samples in Panel F should also be either in Panel F or in the caption.

Our response: Information has been added to the figure caption.

Edits in the manuscript:

I. 924-929: “Horizontal lines within panels A-E display daily mean of respective deployment (with n = 13 for *P. crispera*, n = 7 for *P. oceanica*, n = 6 for reference habitat for O₂ concentration, pH and chlorophyll *a* concentration, and with n = 10 for *P. crispera*, n = 6 in *P. oceanica*, n = 4 in reference habitat for light intensity and temperature) of respective parameters in each habitat. Note for panel F: different letters above box plots indicate significant differences between habitats (ANOVA and subsequent Tukey HSD test), mean of respective deployment with n = 9 for *P. crispera*, n = 4 for *P. oceanica*, n = 2 for reference habitat.”

Reviewer #2 (Remarks to the Author):

Thank you for revisions. I'm satisfied with the work done here, and have just a few minor additional corrections to note:

Our response: We thank the reviewer for the helpful and constructive feedback. We have addressed the remaining points raised by the reviewer (see below).

Lines 59-62: These two sentences are essentially stating the same things – can they be combined?

Our response: The two sentences have been merged.

Edits in the manuscript:

I. 59-61: “In the Mediterranean Sea, rocky hard-bottom communities and commonly identified biodiversity hotspots such as seagrass meadows are declining primarily due to environmental pressures^{5,6,19,20}.”

Line 83: “in the Atlantic”

Our response: Has been edited accordingly.

Lines 99-102: I would remove the second reason here and mention it after the others, not as a reason, but as a consequence of choosing to focus on sessile fauna.

Our response: Has been edited accordingly.

Edits in the manuscript:

I. 102-107: “This invertebrate diversity is likely linked to different habitat characteristics such as micro-niches caused by varying influences on key environmental parameters and ecosystem engineering functions³³. Secondly, the presence of sessile invertebrates potentially reflects the stability and longevity of the red algae mats as habitats⁴⁷. Hence, a particular sampling procedure was chosen to ensure the complete retrieval of sessile invertebrates causing a potential loss of mobile invertebrates.”

Line 125 and throughout the manuscript: change “indiv. per m2 “ to “ind m-2”

Our response: Has been changed here and throughout the manuscript.